# ADAPTIVE GRADIENT METHODS WITH DYNAMIC BOUND OF LEARNING RATE

**Liangchen Luo**[†,*] **Yuanhao Xiong**[‡,*] **Yan Liu**[§], **Xu Sun**[†¶]
[†]MOE Key Lab of Computational Linguistics, School of EECS, Peking University
[‡]College of Information Science and Electronic Engineering, Zhejiang University
[§]Department of Computer Science, University of Southern California
[¶]Center for Data Science, Beijing Institute of Big Data Research, Peking University
[†]{luolc,xusun}@pku.edu.cn [‡]xiongyh@zju.edu.cn [§]yanliu.cs@usc.edu

## ABSTRACT

Adaptive optimization methods such as ADAGRAD, RMSPROP and ADAM have been proposed to achieve a rapid training process with an element-wise scaling term on learning rates. Though prevailing, they are observed to generalize poorly compared with SGD or even fail to converge due to unstable and extreme learning rates. Recent work has put forward some algorithms such as AMSGRAD to tackle this issue but they failed to achieve considerable improvement over existing methods. In our paper, we demonstrate that extreme learning rates can lead to poor performance. We provide new variants of ADAM and AMSGRAD, called ADABOUND and AMSBOUND respectively, which employ dynamic bounds on learning rates to achieve a gradual and smooth transition from adaptive methods to SGD and give a theoretical proof of convergence. We further conduct experiments on various popular tasks and models, which is often insufficient in previous work. Experimental results show that new variants can eliminate the generalization gap between adaptive methods and SGD and maintain higher learning speed early in training at the same time. Moreover, they can bring significant improvement over their prototypes, especially on complex deep networks. The implementation of the algorithm can be found at https://github.com/Luolc/AdaBound.

## 1 INTRODUCTION

There has been tremendous progress in first-order optimization algorithms for training deep neural networks. One of the most dominant algorithms is stochastic gradient descent (SGD) (Robbins & Monro, 1951), which performs well across many applications in spite of its simplicity. However, there is a disadvantage of SGD that it scales the gradient uniformly in all directions. This may lead to poor performance as well as limited training speed when the training data are sparse. To address this problem, recent work has proposed a variety of *adaptive* methods that scale the gradient by square roots of some form of the average of the squared values of past gradients. Examples of such methods include ADAM (Kingma & Lei Ba, 2015), ADAGRAD (Duchi et al., 2011) and RMSPROP (Tieleman & Hinton, 2012). ADAM in particular has become the default algorithm leveraged across many deep learning frameworks due to its rapid training speed (Wilson et al., 2017).

Despite their popularity, the generalization ability and out-of-sample behavior of these adaptive methods are likely worse than their non-adaptive counterparts. Adaptive methods often display faster progress in the initial portion of the training, but their performance quickly plateaus on the unseen data (development/test set) (Wilson et al., 2017). Indeed, the optimizer is chosen as SGD (or with momentum) in several recent state-of-the-art works in natural language processing and computer vision (Luo et al., 2019; Wu & He, 2018), wherein these instances SGD does perform better than adaptive methods. Reddi et al. (2018) have recently proposed a variant of ADAM called AMSGRAD, hoping to solve this problem. The authors provide a theoretical guarantee of convergence but only illustrate its better performance on training data. However, the generalization ability of AMSGRAD

---

[*]Equal contribution. This work was done when the first and second authors were on an internship at DiDi AI Labs.

on unseen data is found to be similar to that of ADAM while a considerable performance gap still exists between AMSGRAD and SGD (Keskar & Socher, 2017; Chen et al., 2018).

In this paper, we first conduct an empirical study on ADAM and illustrate that both extremely large and small learning rates exist by the end of training. The results correspond with the perspective pointed out by Wilson et al. (2017) that the lack of generalization performance of adaptive methods may stem from unstable and extreme learning rates. In fact, introducing non-increasing learning rates, the key point in AMSGRAD, may help abate the impact of huge learning rates, while it neglects possible effects of small ones. We further provide an example of a simple convex optimization problem to elucidate how tiny learning rates of adaptive methods can lead to undesirable non-convergence. In such settings, RMSPROP and ADAM provably do not converge to an optimal solution, and furthermore, however large the initial step size $\alpha$ is, it is impossible for ADAM to fight against the scale-down term.

Based on the above analysis, we propose new variants of ADAM and AMSGRAD, named AD-ABOUND and AMSBOUND, which do not suffer from the negative impact of extreme learning rates. We employ dynamic bounds on learning rates in these adaptive methods, where the lower and upper bound are initialized as zero and infinity respectively, and they both smoothly converge to a constant final step size. The new variants can be regarded as adaptive methods at the beginning of training, and they gradually and smoothly transform to SGD (or with momentum) as time step increases. In this framework, we can enjoy a rapid initial training process as well as good final generalization ability. We provide a convergence analysis for the new variants in the convex setting.

We finally turn to an empirical study of the proposed methods on various popular tasks and models in computer vision and natural language processing. Experimental results demonstrate that our methods have higher learning speed early in training and in the meantime guarantee strong generalization performance compared to several adaptive and non-adaptive methods. Moreover, they can bring considerable improvement over their prototypes especially on complex deep networks.

## 2 NOTATIONS AND PRELIMINARIES

**Notations**    Given a vector $\theta \in \mathbb{R}^d$ we denote its $i$-th coordinate by $\theta_i$; we use $\theta^k$ to denote element-wise power of $k$ and $\|\theta\|$ to denote its $\ell_2$-norm; for a vector $\theta_t$ in the $t$-th iteration, the $i$-th coordinate of $\theta_t$ is denoted as $\theta_{t,i}$ by adding a subscript $i$. Given two vectors $v, w \in \mathbb{R}^d$, we use $\langle v, w \rangle$ to denote their inner product, $v \odot w$ to denote element-wise product, $v/w$ to denote element-wise division, $\max(v, w)$ to denote element-wise maximum and $\min(v, w)$ to denote element-wise minimum. We use $\mathcal{S}_+^d$ to denote the set of all positive definite $d \times d$ matrices. For a vector $a \in \mathbb{R}^d$ and a positive definite matrix $M \in \mathbb{R}^{d \times d}$, we use $a/M$ to denote $M^{-1}a$ and $\sqrt{M}$ to denote $M^{1/2}$. The projection operation $\Pi_{\mathcal{F},M}(y)$ for $M \in \mathcal{S}_+^d$ is defined as $\arg\min_{x \in \mathcal{F}} \|M^{1/2}(x - y)\|$ for $y \in \mathbb{R}^d$. We say $\mathcal{F}$ has bounded diameter $D_\infty$ if $\|x - y\|_\infty \leq D_\infty$ for all $x, y \in \mathcal{F}$.

**Online convex programming**    A flexible framework to analyze iterative optimization methods is the *online optimization problem*. It can be formulated as a repeated game between a player (the algorithm) and an adversary. At step $t$, the algorithm chooses an decision $x_t \in \mathcal{F}$, where $\mathcal{F} \subset \mathbb{R}^d$ is a convex feasible set. Then the adversary chooses a convex loss function $f_t$ and the algorithm incurs loss $f_t(x_t)$. The difference between the total loss $\sum_{t=1}^{T} f_t(x_t)$ and its minimum value for a fixed decision is known as the *regret*, which is represented by $R_T = \sum_{t=1}^{T} f_t(x_t) - \min_{x \in \mathcal{F}} \sum_{t=1}^{T} f_t(x)$. Throughout this paper, we assume that the feasible set $\mathcal{F}$ has bounded diameter and $\|\nabla f_t(x)\|_\infty$ is bounded for all $t \in [T]$ and $x \in \mathcal{F}$. We are interested in algorithms with little regret. Formally speaking, our aim is to devise an algorithm that ensures $R_T = o(T)$, which implies that on average, the model's performance converges to the optimal one. It has been pointed out that an online optimization algorithm with vanishing average regret yields a corresponding stochastic optimization algorithm (Cesa-Bianchi et al., 2002). Thus, following Reddi et al. (2018), we use online gradient descent and stochastic gradient descent synonymously.

**A generic overview of optimization methods**    We follow Reddi et al. (2018) to provide a generic framework of optimization methods in Algorithm 1 that encapsulates many popular adaptive and non-adaptive methods. This is useful for understanding the properties of different optimization methods. Note that the algorithm is still abstract since the functions $\phi_t : \mathcal{F}^t \to \mathbb{R}^d$ and $\psi_t : \mathcal{F}^d \to \mathcal{S}_+^d$ have not been specified. In this paper, we refer to $\alpha$ as initial step size and $\alpha_t/\sqrt{V_t}$ as

---

**Algorithm 1** Generic framework of optimization methods

---

**Input:** $x_1 \in \mathcal{F}$, initial step size $\alpha$, sequence of functions $\{\phi_t, \psi_t\}_{t=1}^{T}$
1: **for** $t = 1$ **to** $T$ **do**
2:     $g_t = \nabla f_t(x_t)$
3:     $m_t = \phi_t(g_1, \cdots, g_t)$ and $V_t = \psi_t(g_1, \cdots, g_t)$
4:     $\alpha_t = \alpha/\sqrt{t}$
5:     $\hat{x}_{t+1} = x_t - \alpha_t m_t/\sqrt{V_t}$
6:     $x_{t+1} = \Pi_{\mathcal{F}, \sqrt{V_t}}(\hat{x}_{t+1})$
7: **end for**

---

learning rate of the algorithm. Note that we employ a design of decreasing step size by $\alpha_t = \alpha/\sqrt{t}$ for it is required for theoretical proof of convergence. However such an aggressive decay of step size typically translates into poor empirical performance, while a simple constant step size $\alpha_t = \alpha$ usually works well in practice. For the sake of clarity, we will use the decreasing scheme for theoretical analysis and the constant schemem for empirical study in the rest of the paper.

Under such a framework, we can summarize the popular optimization methods in Table 1.[1] A few remarks are in order. We can see the scaling term $\psi_t$ is $\mathbb{I}$ in SGD(M), while adaptive methods introduce different kinds of averaging of the squared values of past gradients. ADAM and RMSPROP can be seen as variants of ADAGRAD, where the former ones use an exponential moving average as function $\psi_t$ instead of the simple average used in ADAGRAD. In particular, RMSPROP is essentially a special case of ADAM with $\beta_1 = 0$. AMSGRAD is not listed in the table as it does not has a simple expression of $\psi_t$. It can be defined as $\psi_t = \text{diag}(\hat{v}_t)$ where $\hat{v}_t$ is obtained by the following recursion: $v_t = \beta_2 v_{t-1} + (1 - \beta_2)g_t^2$ and $\hat{v}_t = \max(\hat{v}_{t-1}, v_t)$ with $\hat{v}_0 = v_0 = \mathbf{0}$. The definition of $\phi_t$ is same with that of ADAM. In the rest of the paper we will mainly focus on ADAM due to its generality but our arguments also apply to other similar adaptive methods such as RMSPROP and AMSGRAD.

Table 1: An overview of popular optimization methods using the generic framework.

| | SGD | SGDM | ADAGRAD | RMSPROP | ADAM |
|---|---|---|---|---|---|
| $\phi_t$ | $g_t$ | $\sum\limits_{i=1}^{t} \gamma^{t-i} g_i$ | $g_t$ | $g_t$ | $(1-\beta_1) \sum\limits_{i=1}^{t} \beta_1^{t-i} g_i$ |
| $\psi_t$ | $\mathbb{I}$ | $\mathbb{I}$ | $\text{diag}(\sum\limits_{i=1}^{t} g_i^2)/t$ | $(1-\beta_2)\text{diag}(\sum\limits_{i=1}^{t} \beta_2^{t-i} g_i^2)$ | $(1-\beta_2)\text{diag}(\sum\limits_{i=1}^{t} \beta_2^{t-i} g_i^2)$ |

## 3    THE NON-CONVERGENCE CAUSED BY EXTREME LEARNING RATE

In this section, we elaborate the primary defect in current adaptive methods with a preliminary experiment and a rigorous proof. As mentioned above, adaptive methods like ADAM are observed to perform worse than SGD. Reddi et al. (2018) proposed AMSGRAD to solve this problem but recent work has pointed out AMSGRAD does not show evident improvement over ADAM (Keskar & Socher, 2017; Chen et al., 2018). Since AMSGRAD is claimed to have a smaller learning rate compared with ADAM, the authors only consider large learning rates as the cause for bad performance of ADAM. However, small ones might be a pitfall as well. Thus, we speculate both extremely large and small learning rates of ADAM are likely to account for its ordinary generalization ability.

For corroborating our speculation, we sample learning rates of several weights and biases of ResNet-34 on CIFAR-10 using ADAM. Specifically, we randomly select nine $3 \times 3$ convolutional kernels from different layers and the biases in the last linear layer. As parameters of the same layer usually have similar properties, here we only demonstrate learning rates of nine weights sampled from nine kernels respectively and one bias from the last layer by the end of training, and employ a heatmap to visualize them. As shown in Figure 1, we can find that when the model is close to convergence, learning rates are composed of tiny ones less than 0.01 as well as huge ones greater than 1000.

---

[1]We ignore the debiasing term used in the original version of ADAM in Kingma & Lei Ba (2015) for simplicity. Our arguments apply to the debiased version as well.

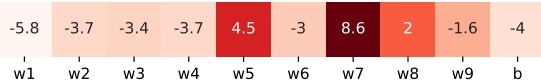

Figure 1: Learning rates of sampled parameters. Each cell contains a value obtained by conducting a logarithmic operation on the learning rate. The lighter cell stands for the smaller learning rate.

The above analysis and observation show that there are indeed learning rates which are too large or too small in the final stage of the training process. AMSGRAD may help abate the impact of huge learning rates, but it neglects the other side of the coin. Insofar, we still have the following two doubts. First, does the tiny learning rate really do harm to the convergence of ADAM? Second, as the learning rate highly depends on the initial step size, can we use a relatively larger initial step size $\alpha$ to get rid of too small learning rates?

To answer these questions, we show that undesirable convergence behavior for ADAM and RM-SPROP can be caused by extremely small learning rates, and furthermore, in some cases no matter how large the initial step size $\alpha$ is, ADAM will still fail to find the right path and converge to some highly suboptimal points. Consider the following sequence of linear functions for $\mathcal{F} = [-1, 1]$:

$$
f_t(x) = \begin{cases} -x, & \text{for } t \bmod C = 1; \\ 2x, & \text{for } t \bmod C = 2; \\ 0, & \text{otherwise} \end{cases}
$$

where $C \in \mathbb{N}$ satisfies: $5\beta_2^{C-2} \leq (1 - \beta_2)/2(4 - \beta_2)$. For this function sequence, it is easy to see that the point $x = -1$ provides the minimum regret. Supposing $\beta_1 = 0$, we show that ADAM converges to a highly suboptimal solution of $x \geq 0$ for this setting. Intuitively, the reasoning is as follows. The algorithm obtains a gradient $-1$ once every $C$ steps, which moves the algorithm in the wrong direction. Then, at the next step it observes a gradient $2$. But the larger gradient $2$ is unable to counteract the effect to wrong direction since the learning rate at this step is scaled down to a value much less than the previous one, and hence $x$ becomes larger and larger as the time step increases. We formalize this intuition in the result below.

**Theorem 1.** *There is an online convex optimization problem where for any initial step size $\alpha$, ADAM has non-zero average regret i.e., $R_T/T \nrightarrow 0$ as $T \to \infty$.*

We relegate all proofs to the appendix. Note that the above example also holds for constant step size $\alpha_t = \alpha$. Also note that vanilla SGD does not suffer from this problem. There is a wide range of valid choices of initial step size $\alpha$ where the average regret of SGD asymptotically goes to $0$, in other words, converges to the optimal solution. This problem can be more obvious in the later stage of a training process in practice when the algorithm gets stuck in some suboptimal points. In such cases, gradients at most steps are close to $0$ and the average of the second order momentum may be highly various due to the property of exponential moving average. Therefore, "correct" signals which appear with a relatively low frequency (i.e. gradient $2$ every $C$ steps in the above example) may not be able to lead the algorithm to a right path, if they come after some "wrong" signals (i.e. gradient $1$ in the example), even though the correct ones have larger absolute value of gradients.

One may wonder if using large $\beta_1$ helps as we usually use $\beta_1$ close to $1$ in practice. However, the following result shows that for any constant $\beta_1$ and $\beta_2$ with $\beta_1 < \sqrt{\beta_2}$, there exists an example where ADAM has non-zero average regret asymptotically regardless of the initial step size $\alpha$.

**Theorem 2.** *For any constant $\beta_1, \beta_2 \in [0, 1)$ such that $\beta_1 < \sqrt{\beta_2}$, there is an online convex optimization problem where for any initial step size $\alpha$, ADAM has non-zero average regret i.e., $R_T/T \nrightarrow 0$ as $T \to \infty$.*

Furthermore, a stronger result stands in the easier stochastic optimization setting.

**Theorem 3.** *For any constant $\beta_1, \beta_2 \in [0, 1)$ such that $\beta_1 < \sqrt{\beta_2}$, there is a stochastic convex optimization problem where for any initial step size $\alpha$, ADAM does not converge to the optimal solution.*

*Remark.* The analysis of ADAM in Kingma & Lei Ba (2015) relies on decreasing $\beta_1$ over time, while here we use constant $\beta_1$. Indeed, since the critical parameter is $\beta_2$ rather than $\beta_1$ in our analysis, it is quite easy to extend our examples to the case using decreasing scheme of $\beta_1$.

As mentioned by Reddi et al. (2018), the condition $\beta_1 < \sqrt{\beta_2}$ is benign and is typically satisfied in the parameter settings used in practice. Such condition is also assumed in convergence proof of Kingma & Lei Ba (2015). The above results illustrate the potential bad impact of extreme learning rates and algorithms are unlikely to achieve good generalization ability without solving this problem.

## 4 ADAPTIVE MOMENT ESTIMATION WITH DYNAMIC BOUND

In this section we develop new variants of optimization methods and provide their convergence analysis. Our aim is to devise a strategy that combines the benefits of adaptive methods, viz. fast initial progress, and the good final generalization properties of SGD. Intuitively, we would like to construct an algorithm that behaves like adaptive methods early in training and like SGD at the end.

---

**Algorithm 2** ADABOUND

**Input:** $x_1 \in \mathcal{F}$, initial step size $\alpha$, $\{\beta_{1t}\}_{t=1}^T$, $\beta_2$, lower bound function $\eta_l$, upper bound function $\eta_u$
1: Set $m_0 = 0$, $v_0 = 0$
2: **for** $t = 1$ **to** $T$ **do**
3:     $g_t = \nabla f_t(x_t)$
4:     $m_t = \beta_{1t} m_{t-1} + (1 - \beta_{1t}) g_t$
5:     $v_t = \beta_2 v_{t-1} + (1 - \beta_2) g_t^2$ and $V_t = \text{diag}(v_t)$
6:     $\hat{\eta}_t = \text{Clip}(\alpha/\sqrt{V_t}, \eta_l(t), \eta_u(t))$ and $\eta_t = \hat{\eta}_t/\sqrt{t}$
7:     $x_{t+1} = \Pi_{\mathcal{F}, \text{diag}(\eta_t^{-1})}(x_t - \eta_t \odot m_t)$
8: **end for**

---

Inspired by *gradient clipping*, a popular technique used in practice that clips the gradients larger than a threshold to avoid gradient explosion, we employ clipping on learning rates in ADAM to propose ADABOUND in Algorithm 2. Consider applying the following operation in ADAM

$$\text{Clip}(\alpha/\sqrt{V_t}, \eta_l, \eta_u),$$

which clips the learning rate element-wisely such that the output is constrained to be in $[\eta_l, \eta_u]$.[2] It follows that SGD(M) with $\alpha = \alpha^*$ can be considered as the case where $\eta_l = \eta_u = \alpha^*$. As for ADAM, $\eta_l = 0$ and $\eta_u = \infty$. Now we can provide the new strategy with the following steps. We employ $\eta_l$ and $\eta_u$ as functions of $t$ instead of constant lower and upper bound, where $\eta_l(t)$ is a non-decreasing function that starts from 0 as $t = 0$ and converges to $\alpha^*$ asymptotically; and $\eta_u(t)$ is a non-increasing function that starts from $\infty$ as $t = 0$ and also converges to $\alpha^*$ asymptotically. In this setting, ADABOUND behaves just like ADAM at the beginning as the bounds have very little impact on learning rates, and it gradually transforms to SGD(M) as the bounds become more and more restricted. We prove the following key result for ADABOUND.

**Theorem 4.** *Let $\{x_t\}$ and $\{v_t\}$ be the sequences obtained from Algorithm 2, $\beta_1 = \beta_{11}$, $\beta_{1t} \leq \beta_1$ for all $t \in [T]$ and $\beta_1/\sqrt{\beta_2} < 1$. Suppose $\eta_l(t+1) \geq \eta_l(t) > 0$, $\eta_u(t+1) \leq \eta_u(t)$, $\eta_l(t) \to \alpha^*$ as $t \to \infty$, $\eta_u(t) \to \alpha^*$ as $t \to \infty$, $L_\infty = \eta_l(1)$ and $R_\infty = \eta_u(1)$. Assume that $\|x - y\|_\infty \leq D_\infty$ for all $x, y \in \mathcal{F}$ and $\|\nabla f_t(x)\| \leq G_2$ for all $t \in [T]$ and $x \in \mathcal{F}$. For $x_t$ generated using the ADABOUND algorithm, we have the following bound on the regret*

$$R_T \leq \frac{D_\infty^2 \sqrt{T}}{2(1-\beta_1)} \sum_{i=1}^d \hat{\eta}_{T,i}^{-1} + \frac{D_\infty^2}{2(1-\beta_1)} \sum_{t=1}^T \sum_{i=1}^d \beta_{1t} \eta_{t,i}^{-1} + (2\sqrt{T} - 1)\frac{R_\infty G_2^2}{1-\beta_1}.$$

The following result falls as an immediate corollary of the above result.

**Corollary 4.1.** *Suppose $\beta_{1t} = \beta_1 \lambda^{t-1}$ in Theorem 4, we have*

$$R_T \leq \frac{D_\infty^2 \sqrt{T}}{2(1-\beta_1)} \sum_{i=1}^d \hat{\eta}_{T,i}^{-1} + \frac{\beta_1 d D_\infty^2}{2(1-\beta_1)(1-\lambda)^2 L_\infty} + (2\sqrt{T} - 1)\frac{R_\infty G_2^2}{1-\beta_1}.$$

---

[2]Here we use constant step size for simplicity. The case using a decreasing scheme of step size is similar. This technique was also mentioned in previous work (Keskar & Socher, 2017).

It is easy to see that the regret of ADABOUND is upper bounded by $O(\sqrt{T})$. Similar to Reddi et al. (2018), one can use a much more modest momentum decay of $\beta_{1t} = \beta_1/t$ and still ensure a regret of $O(\sqrt{T})$. It should be mentioned that one can also incorporate the dynamic bound in AMSGRAD. The resulting algorithm, namely AMSBOUND, also holds a regret of $O(\sqrt{T})$ and the proof of convergence is almost same to Theorem 4 (see Appendix F for details). In next section we will see that AMSBOUND has similar performance to ADABOUND in several well-known tasks.

We end this section with a comparison to the previous work. For the idea of transforming ADAM to SGD, there is a similar work by Keskar & Socher (2017). The authors propose a measure that uses ADAM at first and switches the algorithm to SGD at some specific step. Compared with their approach, our methods have two advantages. First, whether there exists a fixed turning point to distinguish ADAM and SGD is uncertain. So we address this problem with a continuous transforming procedure rather than a "hard" switch. Second, they introduce an extra hyperparameter to decide the switching time, which is not very easy to fine-tune. As for our methods, the flexible parts introduced are two bound functions. We conduct an empirical study of the impact of different kinds of bound functions. The results are placed in Appendix G for we find that the convergence target $\alpha^*$ and convergence speed are not very important to the final results. For the sake of clarity, we will use $\eta_l(t) = 0.1 - \frac{0.1}{(1-\beta_2)t+1}$ and $\eta_u(t) = 0.1 + \frac{0.1}{(1-\beta_2)t}$ in the rest of the paper unless otherwise specified.

## 5 EXPERIMENTS

In this section, we turn to an empirical study of different models to compare new variants with popular optimization methods including SGD(M), ADAGRAD, ADAM, and AMSGRAD. We focus on three tasks: the MNIST image classification task (Lecun et al., 1998), the CIFAR-10 image classification task (Krizhevsky & Hinton, 2009), and the language modeling task on Penn Treebank (Marcus et al., 1993). We choose them due to their broad importance and availability of their architectures for reproducibility. The setup for each task is detailed in Table 2. We run each experiment three times with the specified initialization method from random starting points. A fixed budget on the number of epochs is assigned for training and the decay strategy is introduced in following parts. We choose the settings that achieve the lowest training loss at the end.

Table 2: Summaries of the models utilized for our experiments.

| Dataset | Network Type | Architecture |
|---------|-------------|--------------|
| MNIST | Feedforward | 1-Layer Perceptron |
| CIFAR-10 | Deep Convolutional | DenseNet-121 |
| CIFAR-10 | Deep Convolutional | ResNet-34 |
| Penn Treebank | Recurrent | 1-Layer LSTM |
| Penn Treebank | Recurrent | 2-Layer LSTM |
| Penn Treebank | Recurrent | 3-Layer LSTM |

### 5.1 HYPERPARAMETER TUNING

Optimization hyperparameters can exert great impact on ultimate solutions found by optimization algorithms so here we describe how we tune them. To tune the step size, we follow the method in Wilson et al. (2017). We implement a logarithmically-spaced grid of five step sizes. If the best performing parameter is at one of the extremes of the grid, we will try new grid points so that the best performing parameters are at one of the middle points in the grid. Specifically, we tune over hyperparameters in the following way.

**SGD(M)** For tuning the step size of SGD(M), we first coarsely tune the step size on a logarithmic scale from $\{100, 10, 1, 0.1, 0.01\}$ and then fine-tune it. Whether the momentum is used depends on the specific model but we set the momentum parameter to default value 0.9 for all our experiments. We find this strategy effective given the vastly different scales of learning rates needed for different modalities. For instance, SGD with $\alpha = 10$ performs best for language modeling on PTB but for the ResNet-34 architecture on CIFAR-10, a learning rate of 0.1 for SGD is necessary.

**ADAGRAD**   The initial set of step sizes used for ADAGRAD are: $\{5e\text{-}2, 1e\text{-}2, 5e\text{-}3, 1e\text{-}3, 5e\text{-}4\}$. For the initial accumulator value, we choose the recommended value as 0.

**ADAM & AMSGRAD**   We employ the same hyperparameters for these two methods. The initial step sizes are chosen from: $\{1e\text{-}2, 5e\text{-}3, 1e\text{-}3, 5e\text{-}4, 1e\text{-}4\}$. We turn over $\beta_1$ values of $\{0.9, 0.99\}$ and $\beta_2$ values of $\{0.99, 0.999\}$. We use for the perturbation value $\epsilon = 1e\text{-}8$.

**ADABOUND & AMSBOUND**   We directly apply the default hyperparameters for ADAM (a learning rate of $0.001$, $\beta_1 = 0.9$ and $\beta_2 = 0.999$) in our proposed methods.

Note that for other hyperparameters such as batch size, dropout probability, weight decay and so on, we choose them to match the recommendations of the respective base architectures.

## 5.2   FEEDFORWARD NEURAL NETWORK

We train a simple fully connected neural network with one hidden layer for the multiclass classification problem on MNIST dataset. We run 100 epochs and omit the decay scheme for this experiment.

Figure 2 shows the learning curve for each optimization method on both the training and test set. We find that for training, all algorithms can achieve the accuracy approaching 100%. For the test part, SGD performs slightly better than adaptive methods ADAM and AMSGRAD. Our two proposed methods, ADABOUND and AMSBOUND, display slight improvement, but compared with their prototypes there are still visible increases in test accuracy.

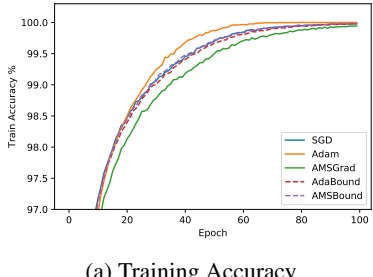 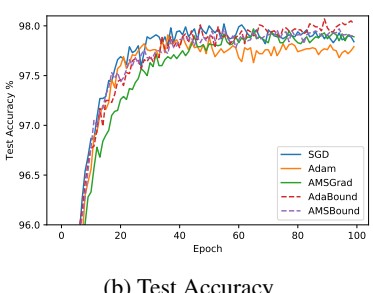

(a) Training Accuracy   (b) Test Accuracy

Figure 2: Training (left) and test accuracy (right) for feedforward neural network on MNIST.

## 5.3   CONVOLUTIONAL NEURAL NETWORK

Using DenseNet-121 (Huang et al., 2017) and ResNet-34 (He et al., 2016), we then consider the task of image classification on the standard CIFAR-10 dataset. In this experiment, we employ the fixed budget of 200 epochs and reduce the learning rates by 10 after 150 epochs.

**DenseNet**   We first run a DenseNet-121 model on CIFAR-10 and our results are shown in Figure 3. We can see that adaptive methods such as ADAGRAD, ADAM and AMSGRAD appear to perform better than the non-adaptive ones early in training. But by epoch 150 when the learning rates are decayed, SGDM begins to outperform those adaptive methods. As for our methods, ADABOUND and AMSBOUND, they converge as fast as adaptive ones and achieve a bit higher accuracy than SGDM on the test set at the end of training. In addition, compared with their prototypes, their performances are enhanced evidently with approximately 2% improvement in the test accuracy.

**ResNet**   Results for this experiment are reported in Figure 3. As is expected, the overall performance of each algorithm on ResNet-34 is similar to that on DenseNet-121. ADABOUND and AMSBOUND even surpass SGDM by 1%. Despite the relative bad generalization ability of adaptive methods, our proposed methods overcome this drawback by allocating bounds for their learning rates and obtain almost the best accuracy on the test set for both DenseNet and ResNet on CIFAR-10.

## 5.4   RECURRENT NEURAL NETWORK

Finally, we conduct an experiment on the language modeling task with Long Short-Term Memory (LSTM) network (Hochreiter & Schmidhuber, 1997). From two experiments above, we observe that

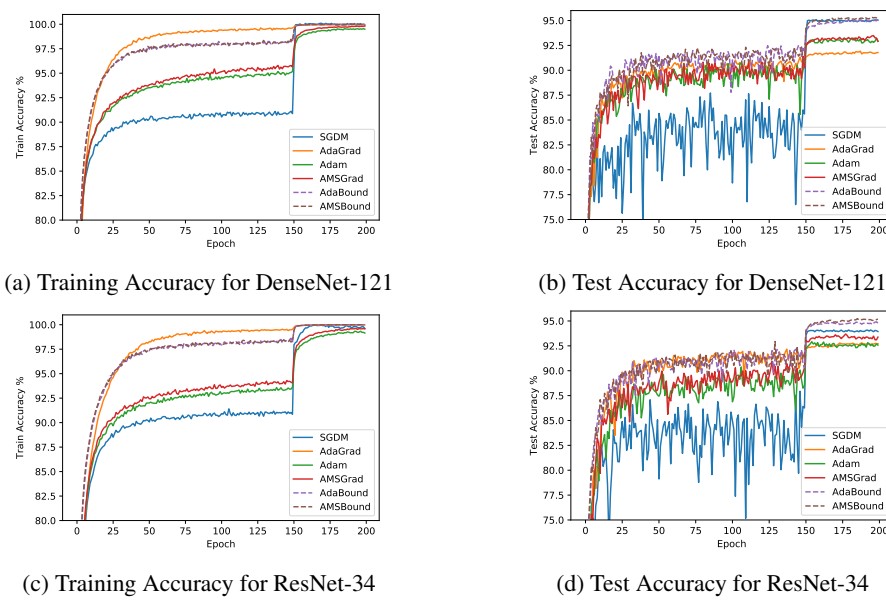

(a) Training Accuracy for DenseNet-121      (b) Test Accuracy for DenseNet-121

(c) Training Accuracy for ResNet-34      (d) Test Accuracy for ResNet-34

Figure 3: Training and test accuracy for DenseNet-121 and ResNet-34 on CIFAR-10.

our methods show much more improvement in deep convolutional neural networks than in perceptrons. Therefore, we suppose that the enhancement is related to the complexity of the architecture and run three models with (**L1**) 1-layer, (**L2**) 2-layer and (**L3**) 3-layer LSTM respectively. We train them on Penn Treebank, running for a fixed budget of 200 epochs. We use perplexity as the metric to evaluate the performance and report results in Figure 4.

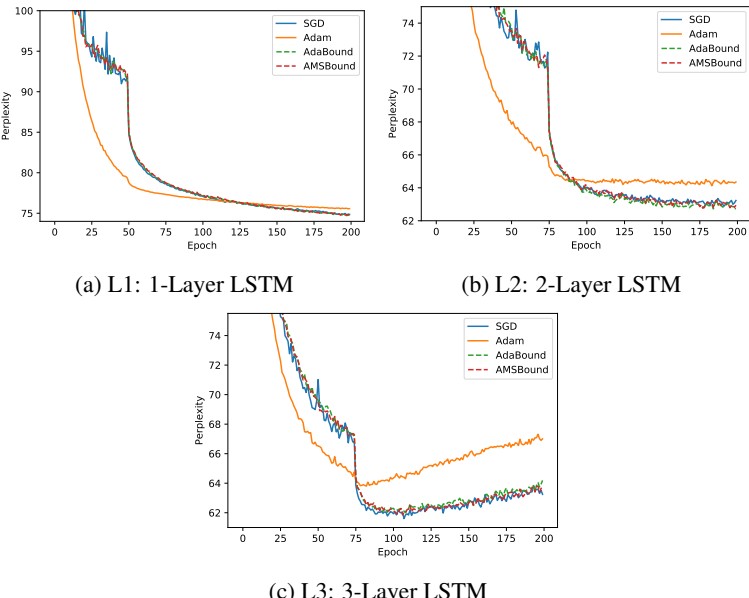

(a) L1: 1-Layer LSTM      (b) L2: 2-Layer LSTM

(c) L3: 3-Layer LSTM

Figure 4: Perplexity curves on the test set comparing SGD, ADAM, ADABOUND and AMSBOUND for the LSTM with different layers on Penn Treebank.

We find that in all models, ADAM has the fastest initial progress but stagnates in worse performance than SGD and our methods. Different from phenomena in previous experiments on the image classification tasks, ADABOUND and AMSBOUND does not display rapid speed at the early training stage but the curves are smoother than that of SGD.

Comparing L1, L2 and L3, we can easily notice a distinct difference of the improvement degree. In L1, the simplest model, our methods perform slightly $1.1\%$ better than ADAM while in L3, the most complex model, they show evident improvement over $2.8\%$ in terms of perplexity. It serves as evidence for the relationship between the model's complexity and the improvement degree.

## 5.5 ANALYSIS

To investigate the efficacy of our proposed algorithms, we select popular tasks from computer vision and natural language processing. Based on results shown above, it is easy to find that ADAM and AMSGRAD usually perform similarly and the latter does not show much improvement for most cases. Their variants, ADABOUND and AMSBOUND, on the other hand, demonstrate a fast speed of convergence compared with SGD while they also exceed two original methods greatly with respect to test accuracy at the end of training. This phenomenon exactly confirms our view mentioned in Section 3 that both large and small learning rates can influence the convergence.

Besides, we implement our experiments on models with different complexities, consisting of a perceptron, two deep convolutional neural networks and a recurrent neural network. The perceptron used on the MNIST is the simplest and our methods perform slightly better than others. As for DenseNet and ResNet, obvious increases in test accuracy can be observed. We attribute this difference to the complexity of the model. Specifically, for deep CNN models, convolutional and fully connected layers play different parts in the task. Also, different convolutional layers are likely to be responsible for different roles (Lee et al., 2009), which may lead to a distinct variation of gradients of parameters. In other words, extreme learning rates (huge or tiny) may appear more frequently in complex models such as ResNet. As our algorithms are proposed to avoid them, the greater enhancement of performance in complex architectures can be explained intuitively. The higher improvement degree on LSTM with more layers on language modeling task also consists with the above analysis.

## 6 FUTURE WORK

Despite superior results of our methods, there still remain several problems to explore. For example, the improvement on simple models are not very inspiring, we can investigate how to achieve higher improvement on such models. Besides, we only discuss reasons for the weak generalization ability of adaptive methods, however, why SGD usually performs well across diverse applications of machine learning still remains uncertain. Last but not least, applying dynamic bounds on learning rates is only one particular way to conduct gradual transformation from adaptive methods to SGD. There might be other ways such as well-designed decay that can also work, which remains to explore.

## 7 CONCLUSION

We investigate existing adaptive algorithms and find that extremely large or small learning rates can result in the poor convergence behavior. A rigorous proof of non-convergence for ADAM is provided to demonstrate the above problem.

Motivated by the strong generalization ability of SGD, we design a strategy to constrain the learning rates of ADAM and AMSGRAD to avoid a violent oscillation. Our proposed algorithms, ADABOUND and AMSBOUND, which employ dynamic bounds on their learning rates, achieve a smooth transition to SGD. They show the great efficacy on several standard benchmarks while maintaining advantageous properties of adaptive methods such as rapid initial progress and hyperparameter insensitivity.

ACKNOWLEDGMENTS

We thank all reviewers for providing the constructive suggestions. We also thank Junyang Lin and Ruixuan Luo for proofreading and doing auxiliary experiments. Xu Sun is the corresponding author of this paper.

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

## APPENDIX

## A    AUXILIARY LEMMAS

**Lemma 1** (Mcmahan & Streeter (2010)). *For any $Q \in \mathcal{S}_+^d$ and convex feasible set $\mathcal{F} \subset \mathbb{R}^d$, suppose $u_1 = \min_{x \in \mathcal{F}} \|Q^{1/2}(x - z_1)\|$ and $u_2 = \min_{x \in \mathcal{F}} \|Q^{1/2}(x - z_2)\|$ then we have $\|Q^{1/2}(u_1 - u_2)\| \leq \|Q^{1/2}(z_1 - z_2)\|$.*

*Proof.* We provide the proof here for completeness. Since $u_1 = \min_{x \in \mathcal{F}} \|Q^{1/2}(x - z_1)\|$ and $u_2 = \min_{x \in \mathcal{F}} \|Q^{1/2}(x - z_2)\|$ and from the property of projection operator we have the following:

$$\langle u_1 - z_1, Q(u_2 - u_1) \rangle \geq 0 \text{ and } \langle u_2 - z_2, Q(u_1 - u_2) \rangle \geq 0.$$

Combining the above inequalities, we have

$$\langle z_2 - z_1, Q(u_2 - u_1) \rangle \geq \langle u_2 - u_1, Q(u_2 - u_1) \rangle. \tag{1}$$

Also, observe the following:

$$\langle z_2 - z_1, Q(u_2 - u_1) \rangle \leq \frac{1}{2} \left[ \langle u_2 - u_1, Q(u_2 - u_1) \rangle + \langle z_2 - z_1, Q(z_2 - z_1) \rangle \right].$$

The above inequality can be obtained from the fact that

$$\langle (u_2 - u_1) - (z_2 - z_1), Q((u_2 - u_1) - (z_2 - z_1)) \rangle \geq 0 \text{ as } Q \in \mathcal{S}_+^d$$

and rearranging the terms. Combining the above inequality with Equation (1), we have the required the result. $\square$

**Lemma 2.** *Suppose $m_t = \beta_1 m_{t-1} + (1 - \beta_1) g_t$ with $m_0 = \mathbf{0}$ and $0 \leq \beta_1 < 1$. We have*

$$\sum_{t=1}^{T} \|m_t\|^2 \leq \sum_{t=1}^{T} \|g_t\|^2.$$

*Proof.* If $\beta_1 = 0$, the equality directly holds due to $m_t = g_t$. Otherwise, $0 < \beta_1 < 1$. For any $\theta > 0$ we have

$$\begin{aligned}
\|m_t\|^2 &= \|\beta_1 m_{t-1}\|^2 + \|(1 - \beta_1) g_t\|^2 + 2\langle \beta_1 m_{t-1}, (1 - \beta_1) g_t \rangle \\
&\leq \|\beta_1 m_{t-1}\|^2 + \|(1 - \beta_1) g_t\|^2 + \theta \|\beta_1 m_{t-1}\|^2 + 1/\theta \|(1 - \beta_1) g_t\|^2 \\
&= (1 + \theta) \|\beta_1 m_{t-1}\|^2 + (1 + 1/\theta) \|(1 - \beta_1) g_t\|^2
\end{aligned}$$

The inequality follows from Cauchy–Schwarz and Young's inequality. In particular, let $\theta = 1/\beta_1 - 1$. Then we have

$$\|m_t\|^2 \leq \beta_1 \|m_{t-1}\|^2 + (1 - \beta_1) \|g_t\|^2.$$

Dividing both sides by $\beta_1^t$, we get

$$\frac{\|m_t\|^2}{\beta_1^t} \leq \frac{\|m_{t-1}\|^2}{\beta_1^{t-1}} + \frac{(1 - \beta_1) \|g_t\|^2}{\beta_1^t}.$$

Note that $m_0 = \mathbf{0}$. Hence,

$$\frac{\|m_t\|^2}{\beta_1^t} \leq (1 - \beta_1) \sum_{i=1}^{t} \|g_i\|^2 \beta_1^{-i}.$$

Then multiplying both sides by $\beta_1^t$ we obtain

$$\|m_t\|^2 \leq (1 - \beta_1) \sum_{i=1}^{t} \|g_i\|^2 \beta_1^{t-i}.$$

Take the summation of above inequality over $t = 1, 2, \cdots, T$, we have

$$
\sum_{t=1}^{T} \|m_t\|^2 \leq (1 - \beta_1) \sum_{t=1}^{T} \sum_{i=1}^{t} \|g_i\|^2 \beta_1^{t-i}
$$

$$
= (1 - \beta_1) \sum_{i=1}^{T} \sum_{t=i}^{T} \|g_i\|^2 \beta_1^{t-i}
$$

$$
\leq \sum_{t=1}^{T} \|g_t\|^2.
$$

The second inequality is due to the following fact of geometric series

$$
\sum_{i=0}^{N} \beta_1^i \leq \sum_{i=0}^{\infty} \beta_1^i = \frac{1}{1 - \beta_1}, \text{ for } 0 < \beta_1 < 1.
$$

We complete the proof.

$\square$

## B  PROOF OF THEOREM 1

*Proof.* First, we rewrite the update of ADAM in Algorithm 1 in the following recursion form:

$$
m_{t,i} = \beta_1 m_{t-1,i} + (1 - \beta_1) g_{t,i} \text{ and } v_{t,i} = \beta_2 v_{t-1,i} + (1 - \beta_2) g_{t,i}^2 \tag{2}
$$

where $m_{0,i} = 0$ and $v_{0,i} = 0$ for all $i \in [d]$ and $\psi_t = \text{diag}(v_t)$. We consider the setting where $f_t$ are linear functions and $\mathcal{F} = [-1, 1]$. In particular, we define the following function sequence:

$$
f_t(x) = \begin{cases} -x, & \text{for } t \bmod C = 1; \\ 2x, & \text{for } t \bmod C = 2; \\ 0, & \text{otherwise} \end{cases}
$$

where $C \in \mathbb{N}$ satisfies the following:

$$
5\beta_2^{C-2} \leq \frac{1}{2} \cdot \frac{1 - \beta_2}{4 - \beta_2}. \tag{3}
$$

It is not hard to see that the condition hold for large constant $C$ that depends on $\beta_2$.

Since the problem is one-dimensional, we drop indices representing coordinates from all quantities in Algorithm 1. For this function sequence, it is easy to see that the point $x = -1$ provides the minimum regret. Consider the execution of ADAM algorithm for this sequence of functions with $\beta_1 = 0$. Note that since gradients of these functions are bounded, $\mathcal{F}$ has bounded $D_\infty$ diameter and $\beta_1^2/\beta_2 < 1$ as $\beta_1 = 0$, the conditions on the parameters required for ADAM are satisfied (Kingma & Lei Ba, 2015). The gradients have the following form:

$$
\nabla f_i(x) = \begin{cases} -1, & \text{for } i \bmod C = 1; \\ 2, & \text{for } i \bmod C = 2; \\ 0, & \text{otherwise.} \end{cases}
$$

Let $\tau \in \mathbb{N}, \tau > 1$ be such that

$$
\frac{\alpha}{\sqrt{Ct + 1}} \frac{1}{\sqrt{(1 - \beta_2)(\beta_2^C + 4\beta_2^{C-1} + 1)}} \leq 1, \tag{4}
$$

$$
\frac{\alpha}{\sqrt{Ct + 2}} \frac{2}{\sqrt{(1 - \beta_2)(4 + \beta_2)}} \leq 1, \tag{5}
$$

for all $t \geq \tau$. We start with the following preliminary result.

**Lemma 3.** *For the parameter settings and conditions assumed in Theorem 1, there is a $t' \geq \tau$ such that $x_{Ct'+1} \geq 0$.*

*Proof by contradiction.* Assume that $x_{Ct+1} < 0$ for all $t \geq \tau$. Firstly, for $t \geq \tau$, we observe the following inequalities:

$$
\begin{aligned}
v_{Ct+1} &= \beta_2 v_{Ct} + (1 - \beta_2) \\
&= (1 - \beta_2)(1 + \sum_{i=1}^{t} \beta_2^{Ci} + 4 \sum_{i=1}^{t} \beta_2^{Ci-1}) \\
&\geq (1 - \beta_2)(\beta_2^C + 4\beta_2^{C-1} + 1),
\end{aligned}
\tag{6}
$$

$$
\begin{aligned}
v_{Ct+1} &= \beta_2 v_{Ct} + (1 - \beta_2) \\
&= (1 - \beta_2)(\sum_{i=1}^{t} \beta_2^{Ci} + 4 \sum_{i=1}^{t} \beta_2^{Ci-1}) + (1 - \beta_2) \\
&\leq (1 - \beta_2)\frac{\beta_2^C + 4\beta_2^{C-1}}{1 - \beta_2^C} + (1 - \beta_2) \\
&\leq 5\beta_2^{C-1} + (1 - \beta_2) < 9,
\end{aligned}
\tag{7}
$$

$$
\begin{aligned}
v_{Ct+2} &= (1 - \beta_2)(\sum_{i=0}^{t} \beta_2^{Ci+1} + 4 \sum_{i=0}^{t} \beta_2^{Ci}) \\
&= (1 - \beta_2)(4 + \beta_2)\frac{1 - \beta_2^{Ct}}{1 - \beta_2^C} \\
&\leq 4 + \beta_2 < 9.
\end{aligned}
\tag{8}
$$

From the $(C\tau + 1)$-th update of ADAM in Equation (2), we obtain:

$$
\begin{aligned}
\hat{x}_{C\tau+2} &= x_{C\tau+1} + \frac{\alpha}{\sqrt{C\tau + 1}} \frac{1}{\sqrt{v_{C\tau+1}}} \\
&< \frac{\alpha}{\sqrt{C\tau + 1}} \frac{1}{\sqrt{(1 - \beta_2)(\beta_2^C + 4\beta_2^{C-1} + 1)}} \leq 1.
\end{aligned}
$$

The first inequality follows from $x_{Ct+1} < 0$ and Equation (6). The last inequality follows from Equation (4). Therefore, we have $-1 \leq x_{C\tau+1} < \hat{x}_{C\tau+2} < 1$ and hence $x_{C\tau+2} = \hat{x}_{C\tau+2}$. Then after the $(C\tau + 2)$-th update, we have:

$$
\begin{aligned}
\hat{x}_{C\tau+3} &= x_{C\tau+2} - \frac{\alpha}{\sqrt{C\tau + 2}} \frac{2}{\sqrt{v_{C\tau+2}}} \\
&= x_{C\tau+1} + \frac{\alpha}{\sqrt{C\tau + 1}} \frac{1}{\sqrt{v_{C\tau+1}}} - \frac{\alpha}{\sqrt{C\tau + 2}} \frac{2}{\sqrt{v_{C\tau+2}}} \\
&\geq x_{C\tau+1} + \frac{1}{\sqrt{C\tau + C}} \frac{\alpha\beta_2(v_{C\tau+1} - 4v_{C\tau})}{\sqrt{v_{C\tau+1}v_{C\tau+2}}(\sqrt{v_{C\tau+2}} + 2\sqrt{v_{C\tau+1}})} \\
&\geq x_{C\tau+1} + \frac{1}{\sqrt{\tau + 1}} \frac{\alpha\beta_2}{81\sqrt{C}}(v_{C\tau+1} - 4v_{C\tau}) \\
&\geq x_{C\tau+1} + \frac{1}{\sqrt{\tau + 1}} \frac{\alpha\beta_2(1 - \beta_2)}{162\sqrt{C}} \\
&= x_{C\tau+1} + \frac{\kappa}{\sqrt{\tau + 1}},
\end{aligned}
\tag{9}
$$

where $\kappa = \alpha\beta_2(1 - \beta_2)/162\sqrt{C}$ is a constant that depends on $\alpha$, $\beta_2$ and $C$. The first inequality follows from Equation (2). The second inequality follows from Equations (7) and (8). The last

inequality is due to the following lower bound:

$$
\begin{aligned}
v_{Ct+1} - 4v_{Ct} &= \beta_2 v_{Ct} + (1-\beta_2) - 4v_{Ct} \\
&= (4-\beta_2)\left[\frac{1-\beta_2}{4-\beta_2} - v_{Ct}\right] \\
&= (4-\beta_2)\left[\frac{1-\beta_2}{4-\beta_2} - (1-\beta_2)(\sum_{i=1}^{t}\beta_2^{Ci-1} + 4\sum_{i=1}^{t}\beta_2^{Ci-2})\right] \\
&\geq (4-\beta_2)\left[\frac{1-\beta_2}{4-\beta_2} - (1-\beta_2)(\frac{\beta_2^{C-1}}{1-\beta_2^C} + \frac{4\beta_2^{C-2}}{1-\beta_2^C})\right] \\
&\geq (4-\beta_2)\left[\frac{1-\beta_2}{4-\beta_2} - 5\beta_2^{C-2}\right] \\
&\geq (4-\beta_2)\cdot\frac{1}{2}\cdot\frac{1-\beta_2}{4-\beta_2} \\
&= \frac{1-\beta_2}{2},
\end{aligned}
$$

where the last inequality follows from Equation (3). Therefore, we have $-1 \leq x_{C\tau+1} < \hat{x}_{C\tau+3} < x_{C\tau+2} < 1$. Furthermore, since gradients $\nabla f_i(x) = 0$ when $i \bmod C \neq 1$ or 2, we have

$$
\begin{aligned}
x_{C\tau+4} &= \hat{x}_{C\tau+3} = x_{C\tau+3}, \\
x_{C\tau+5} &= \hat{x}_{C\tau+4} = x_{C\tau+4}, \\
&\cdots \\
x_{C(\tau+1)+1} &= \hat{x}_{C(\tau+1)+1} = x_{C(\tau+1)}.
\end{aligned}
$$

Then, following Equation (9) we have

$$
x_{C(\tau+1)+1} - x_{C\tau+1} \geq \frac{\kappa}{\sqrt{\tau+1}}.
$$

Similarly, we can subsequently obtain

$$
x_{C(\tau+2)+1} - x_{C(\tau+1)+1} \geq \frac{\kappa}{\sqrt{\tau+2}},
$$

and generally

$$
x_{C(t+1)+1} - x_{Ct+1} \geq \frac{\kappa}{\sqrt{t+1}}
$$

for all $t \geq \tau$. Therefore,

$$
\begin{aligned}
x_{Ct+1} &\geq x_{C\tau+1} + \frac{\kappa}{\sqrt{\tau+1}} + \frac{\kappa}{\sqrt{\tau+2}} + \cdots + \frac{\kappa}{\sqrt{t}} \\
&\geq -1 + \kappa\sum_{n=\tau+1}^{t}\frac{1}{\sqrt{n}} \\
&\geq -1 + \kappa\int_{\tau+1}^{t+1}\frac{\mathrm{d}x}{\sqrt{x}} \\
&= -1 + 2\kappa(\sqrt{t+1} - \sqrt{\tau+1})
\end{aligned}
$$

for $t \geq \tau$. Let $t'$ be such that $2\kappa(\sqrt{t'+1} - \sqrt{\tau+1}) \geq 1$, then $x_{Ct'+1} \geq 0$. This contradicts the assumption that $x_{Ct+1} < 0$ for all $t \geq \tau$. We complete the proof of this lemma. □

We now return to the proof of Theorem 1. The following analysis focuses on iterations after $Ct'+1$ such that $x_{Ct'+1} \geq 0$. Note that any regret before $Ct'+1$ is just a constant since $t'$ is independent of $T$ and thus, the average regret is negligible as $T \to \infty$.

Our claim is that, $x_k \geq 0$ for all $k \in \mathbb{N}$, $k \geq Ct'+1$. To prove this, we resort to the principle of mathematical induction. Suppose for some $t \in \mathbb{N}$, $t \geq t'$, we have $x_{Ct+1} \geq 0$. Our aim is to prove that $x_i \geq 0$ for all $i \in \mathbb{N} \cap [Ct+2, C(t+1)+1]$.

From the $(Ct+1)$-th update of ADAM in Equation (2), we obtain:

$$\hat{x}_{Ct+2} = x_{Ct+1} + \frac{\alpha}{\sqrt{Ct+1}} \frac{1}{\sqrt{v_{Ct+1}}} \geq 0.$$

We consider the following two cases:

1. Suppose $\hat{x}_{Ct+2} > 1$, then $x_{Ct+2} = \Pi_{\mathcal{F}}(\hat{x}_{Ct+2}) = \min\{\hat{x}_{Ct+2}, 1\} = 1$ (note that in one-dimension, $\Pi_{\mathcal{F}, \sqrt{V_t}} = \Pi_{\mathcal{F}}$ is the simple Euclidean projection). After the $(Ct+2)$-th update, we have:

$$\hat{x}_{Ct+3} = x_{Ct+2} - \frac{\alpha}{\sqrt{Ct+2}} \frac{2}{\sqrt{v_{Ct+2}}}$$

$$\geq 1 - \frac{\alpha}{\sqrt{Ct+2}} \frac{2}{\sqrt{(1-\beta_2)(4+\beta_2)}} \geq 0.$$

   The last inequality follows from Equation (5). The first inequality follows from

$$v_{Ct+2} = (1-\beta_2)(\sum_{i=0}^{t} \beta_2^{Ci+1} + 4\sum_{i=0}^{t} \beta_2^{Ci}) \geq (1-\beta_2)(4+\beta_2).$$

2. Suppose $\hat{x}_{Ct+2} \leq 1$, then after the $(Ct+2)$-th update, similar to Equation (9), we have:

$$\hat{x}_{Ct+3} \geq x_{Ct+1} + \frac{\kappa}{\sqrt{t+1}} \geq 0.$$

In both cases, $\hat{x}_{Ct+3} \geq 0$, which translates to $x_{Ct+3} = \hat{x}_{Ct+3} \geq 0$. Furthermore, since gradients $\nabla f_i(x) = 0$ when $i \bmod C \neq 1$ or 2, we have

$$x_{Ct+4} = \hat{x}_{Ct+3} = x_{Ct+3} \geq 0,$$
$$x_{Ct+5} = \hat{x}_{Ct+4} = x_{Ct+4} \geq 0,$$
$$\cdots$$
$$x_{C(t+1)+1} = \hat{x}_{C(t+1)+1} = x_{C(t+1)} \geq 0.$$

Therefore, given $x_{Ct'+1} = 0$, it holds for all $k \in \mathbb{N}$, $k \geq Ct' + 1$ by the principle of mathematical induction. Thus, we have

$$\sum_{i=1}^{C} f_{kC+i}(x_{kC+i}) - \sum_{i=1}^{C} f_{kC+i}(-1) \geq 0 - (-1) = 1,$$

where $k \in \mathbb{N}$, $k \geq t'$. Therefore, when $t \geq t'$, for every $C$ steps, ADAM suffers a regret of at least 1. More specifically, $R_T \geq (T-t')/C$. Thus, $R_T/T \nrightarrow 0$ as $T \to \infty$, which completes the proof. $\square$

## C   PROOF OF THEOREM 2

Theorem 2 generalizes the optimization setting used in Theorem 1. We notice that the example proposed by Reddi et al. (2018) in their Appendix B already satisfies the constraints listed in Theorem 2. Here we provide the setting of the example for completeness.

*Proof.* Consider the setting where $f_t$ are linear functions and $\mathcal{F} = [-1, 1]$. In particular, we define the following function sequence:

$$f_t(x) = \begin{cases} Cx, & \text{for } t \bmod C = 1; \\ -x, & \text{otherwise,} \end{cases}$$

where $C \in \mathbb{N}$, $C \bmod 2 = 0$ satisfies the following:

$$(1-\beta_1)\beta_1^{C-1} C \leq 1 - \beta_1^{C-1},$$

$$\beta_2^{(C-2)/2} C^2 \leq 1,$$

$$\frac{3(1-\beta_1)}{2\sqrt{1-\beta_2}}\left(1 + \frac{\gamma(1-\gamma^{C-1})}{1-\gamma}\right) + \frac{\beta_1^{C/2-1}}{1-\beta_1} < \frac{C}{3},$$

where $\gamma = \beta_1/\sqrt{\beta_2} < 1$. It is not hard to see that these conditions hold for large constant $C$ that depends on $\beta_1$ and $\beta_2$. According to the proof given by Reddi et al. (2018) in their Appendix B, in such a setting $R_T/T \nrightarrow 0$ as $T \to \infty$, which completes the proof.

$\square$

## D   PROOF OF THEOREM 3

The example proposed by Reddi et al. (2018) in their Appendix C already satisfies the constraints listed in Theorem 3. Here we provide the setting of the example for completeness.

*Proof.* Let $\delta$ be an arbitrary small positive constant. Consider the following one dimensional stochastic optimization setting over the domain $[-1, 1]$. At each time step $t$, the function $f_t(x)$ is chosen as follows:

$$f_t(x) = \begin{cases} Cx, & \text{with probability } p := \frac{1+\delta}{C+1} \\ -x, & \text{with probability } 1 - p, \end{cases}$$

where $C$ is a large constant that depends on $\beta_1$, $\beta_2$ and $\delta$. The expected function is $F(x) = \delta x$. Thus the optimal point over $[-1, 1]$ is $x^* = -1$. The step taken by ADAM is

$$\Delta_t = \frac{-\alpha_t \left( \beta_1 m_{t-1} + (1 - \beta_1) g_t \right)}{\sqrt{\beta_2 v_{t-1} + (1 - \beta_2) g_t^2}}.$$

According to the proof given by Reddi et al. (2018) in their Appendix C, there exists a large enough $C$ such that $\mathbb{E}[\Delta_t] \geq 0$, which then implies that the ADAM's step keep drifting away from the optimal solution $x^* = -1$. Note that there is no limitation of the initial step size $\alpha$ by now. Therefore, we complete the proof.

$\square$

## E   PROOF OF THEOREM 4

*Proof.* Let $x^* = \arg\min_{x \in \mathcal{F}} \sum_{t=1}^{T} f_t(x)$, which exists since $\mathcal{F}$ is closed and convex. We begin with the following observation:

$$x_{t+1} = \Pi_{\mathcal{F}, \mathrm{diag}(\eta_t^{-1})}(x_t - \eta_t \odot m_t) = \min_{x \in \mathcal{F}} \|\eta_t^{-1/2} \odot (x - (x_t - \eta_t \odot m_t))\|.$$

Using Lemma 1 with $u_1 = x_{t+1}$ and $u_2 = x^*$, we have the following:

$$\begin{aligned}
\|\eta_t^{-1/2} \odot (x_{t+1} - x^*)\|^2 &\leq \|\eta_t^{-1/2} \odot (x_t - \eta_t \odot m_t - x^*)\|^2 \\
&= \|\eta_t^{-1/2} \odot (x_t - x^*)\|^2 + \|\eta_t^{1/2} \odot m_t\|^2 - 2\langle m_t, x_t - x^* \rangle \\
&= \|\eta_t^{-1/2} \odot (x_t - x^*)\|^2 + \|\eta_t^{1/2} \odot m_t\|^2 \\
&\quad - 2\langle \beta_{1t} m_{t-1} + (1 - \beta_{1t}) g_t, x_t - x^* \rangle.
\end{aligned}$$

Rearranging the above inequality, we have

$$\begin{aligned}
\langle g_t, x_t - x^* \rangle &\leq \frac{1}{2(1 - \beta_{1t})} \left[ \|\eta_t^{-1/2} \odot (x_t - x^*)\|^2 - \|\eta_t^{-1/2} \odot (x_{t+1} - x^*)\|^2 \right] \\
&\quad + \frac{1}{2(1 - \beta_{1t})} \|\eta_t^{1/2} \odot m_t\|^2 + \frac{\beta_{1t}}{1 - \beta_{1t}} \langle m_{t-1}, x_t - x^* \rangle \\
&\leq \frac{1}{2(1 - \beta_{1t})} \left[ \|\eta_t^{-1/2} \odot (x_t - x^*)\|^2 - \|\eta_t^{-1/2} \odot (x_{t+1} - x^*)\|^2 \right] \qquad (10) \\
&\quad + \frac{1}{2(1 - \beta_{1t})} \|\eta_t^{1/2} \odot m_t\|^2 + \frac{\beta_{1t}}{2(1 - \beta_{1t})} \|\eta_t^{1/2} \odot m_{t-1}\|^2 \\
&\quad + \frac{\beta_{1t}}{2(1 - \beta_{1t})} \|\eta_t^{-1/2} \odot (x_t - x^*)\|^2.
\end{aligned}$$

The second inequality follows from simple application of Cauchy–Schwarz and Young's inequality. We now use the standard approach of bounding the regret at each step using convexity of the functions $\{f_t\}_{t=1}^T$ in the following manner:

$$
\sum_{t=1}^T f_t(x_t) - f_t(x^*) \leq \sum_{t=1}^T \langle g_t, x_t - x^* \rangle
$$

$$
\leq \sum_{t=1}^T \left[ \frac{1}{2(1-\beta_{1t})} \left[ \|\eta_t^{-1/2} \odot (x_t - x^*)\|^2 - \|\eta_t^{-1/2} \odot (x_{t+1} - x^*)\|^2 \right] \right.
$$
$$
+ \frac{1}{2(1-\beta_{1t})} \|\eta_t^{1/2} \odot m_t\|^2 + \frac{\beta_{1t}}{2(1-\beta_{1t})} \|\eta_t^{1/2} \odot m_{t-1}\|^2
$$
$$
\left. + \frac{\beta_{1t}}{2(1-\beta_{1t})} \|\eta_t^{-1/2} \odot (x_t - x^*)\|^2 \right].
$$
(11)

The first inequality is due to the convexity of functions $\{f_t\}_{t=1}^T$. The second inequality follows from the bound in Equation (10). For further bounding this inequality, we need the following intermedia result.

**Lemma 4.** *For the parameter settings and conditions assumed in Theorem 4, we have*

$$
\sum_{t=1}^T \left[ \frac{1}{2(1-\beta_{1t})} \|\eta_t^{1/2} \odot m_t\|^2 + \frac{\beta_{1t}}{2(1-\beta_{1t})} \|\eta_t^{1/2} \odot m_{t-1}\|^2 \right] \leq (2\sqrt{T}-1)\frac{R_\infty G_2^2}{1-\beta_1}.
$$

*Proof.* By definition of $\eta_t$, we have

$$
L_\infty \leq \sqrt{t}\|\eta_t\|_\infty \leq R_\infty.
$$

Hence,

$$
\sum_{t=1}^T \left[ \frac{1}{2(1-\beta_{1t})} \|\eta_t^{1/2} \odot m_t\|^2 + \frac{\beta_{1t}}{2(1-\beta_{1t})} \|\eta_t^{1/2} \odot m_{t-1}\|^2 \right]
$$

$$
\leq \sum_{t=1}^T \left[ \frac{R_\infty}{2(1-\beta_{1t})\sqrt{t}} \|m_t\|^2 + \frac{\beta_{1t}R_\infty}{2(1-\beta_{1t})\sqrt{t}} \|m_{t-1}\|^2 \right]
$$

$$
\leq \sum_{t=1}^T \left[ \frac{R_\infty}{2(1-\beta_1)\sqrt{t}} \|m_t\|^2 + \frac{R_\infty}{2(1-\beta_1)\sqrt{t}} \|m_{t-1}\|^2 \right]
$$

$$
= \frac{R_\infty}{2(1-\beta_1)} \left[ \sum_{t=1}^T \frac{\|m_t\|^2}{\sqrt{t}} + \sum_{t=1}^T \frac{\|m_{t-1}\|^2}{\sqrt{t}} \right]
$$

$$
\leq \frac{R_\infty}{2(1-\beta_1)} \left[ \frac{1}{T} \left[ \sum_{t=1}^T \|m_t\| t^{-1/4} \right]^2 + \frac{1}{T} \left[ \sum_{t=1}^T \|m_{t-1}\| t^{-1/4} \right]^2 \right]
$$

$$
\leq \frac{R_\infty}{2(1-\beta_1)T} \left[ \sum_{t=1}^T \|m_t\|^2 \cdot \sum_{t=1}^T t^{-1/2} + \sum_{t=1}^T \|m_{t-1}\|^2 \cdot \sum_{t=1}^T t^{-1/2} \right]
$$

$$
\leq \frac{R_\infty G_2^2}{(1-\beta_1)} \sum_{t=1}^T t^{-1/2}
$$

$$
\leq (2\sqrt{T}-1)\frac{R_\infty G_2^2}{(1-\beta_1)}.
$$

The second inequality is due to $\beta_{1t} \leq \beta_1 < 1$. The third inequality follows from Jensen inequality and the fourth inequality follows from Cauchy–Schwarz inequality. The fifth inequality follows from Lemma 2 and $m_0 = 0$. The last inequality is due to the following upper bound:

$$
\sum_{t=1}^T \frac{1}{\sqrt{t}} \leq 1 + \int_{t=1}^T \frac{dt}{\sqrt{t}} = 2\sqrt{T} - 1.
$$

We complete the proof of this lemma. □

We now return to the proof of Theorem 4. Using the above lemma in Equation (11), we have

$$
\sum_{t=1}^{T} f_t(x_t) - f_t(x^*)
$$

$$
\leq \sum_{t=1}^{T} \left[ \frac{1}{2(1-\beta_{1t})} \left[ \|\eta_t^{-1/2} \odot (x_t - x^*)\|^2 - \|\eta_t^{-1/2} \odot (x_{t+1} - x^*)\|^2 \right] \right.
$$

$$
\left. + \frac{\beta_{1t}}{2(1-\beta_{1t})} \|\eta_t^{-1/2} \odot (x_t - x^*)\|^2 \right] + (2\sqrt{T} - 1)\frac{R_\infty G_2^2}{1-\beta_1}
$$

$$
\leq \frac{1}{2(1-\beta_1)} \left[ \|\eta_t^{-1/2} \odot (x_1 - x^*)\|^2 + \sum_{t=2}^{T} \left[ \|\eta_t^{-1/2} \odot (x_t - x^*)\|^2 - \|\eta_{t-1}^{-1/2} \odot (x_t - x^*)\|^2 \right] \right]
$$

$$
+ \sum_{t=1}^{T} \frac{\beta_{1t}}{2(1-\beta_1)} \|\eta_t^{-1/2} \odot (x_t - x^*)\|^2 + (2\sqrt{T} - 1)\frac{R_\infty G_2^2}{1-\beta_1}
$$

$$
= \frac{1}{2(1-\beta_1)} \left[ \sum_{i=1}^{d} \eta_{1,i}^{-1}(x_{1,i} - x_i^*)^2 + \sum_{t=2}^{T}\sum_{i=1}^{d}(x_{t,i} - x_i^*)^2 \left[ \eta_{t,i}^{-1} - \eta_{t-1,i}^{-1} \right] \right.
$$

$$
\left. + \sum_{t=1}^{T}\sum_{i=1}^{d} \beta_{1t}(x_{t,i} - x_i^*)^2 \eta_{t,i}^{-1} \right] + (2\sqrt{T} - 1)\frac{R_\infty G_2^2}{1-\beta_1}.
$$

$$(12)$$

The second inequality use the fact that $\beta_{1t} \leq \beta_1 < 1$. In order to further simplify the bound in Equation (12), we need to use telescopic sum. We observe that, by definition of $\eta_t$, we have

$$
\eta_{t,i}^{-1} \geq \eta_{t-1,i}^{-1}.
$$

Using the $D_\infty$ bound on the feasible region and making use of the above property in Equation (12), we have

$$
\sum_{t=1}^{T} f_t(x_t) - f_t(x^*)
$$

$$
\leq \frac{D_\infty^2}{2(1-\beta_1)} \left[ \sum_{i=1}^{d} \eta_{1,i}^{-1} + \sum_{t=2}^{T}\sum_{i=1}^{d} \left[ \eta_{t,i}^{-1} - \eta_{t-1,i}^{-1} \right] + \sum_{t=1}^{T}\sum_{i=1}^{d} \beta_{1t}\eta_{t,i}^{-1} \right] + (2\sqrt{T} - 1)\frac{R_\infty G_2^2}{1-\beta_1}
$$

$$
= \frac{D_\infty^2 \sqrt{T}}{2(1-\beta_1)} \sum_{i=1}^{d} \hat{\eta}_{T,i}^{-1} + \frac{D_\infty^2}{2(1-\beta_1)} \sum_{t=1}^{T}\sum_{i=1}^{d} \beta_{1t}\eta_{t,i}^{-1} + (2\sqrt{T} - 1)\frac{R_\infty G_2^2}{1-\beta_1}.
$$

The equality follows from simple telescopic sum, which yields the desired result. It is easy to see that the regret of ADABOUND is upper bounded by $O(\sqrt{T})$.

□

## F   AMSBOUND

**Theorem 5.** *Let $\{x_t\}$ and $\{v_t\}$ be the sequences obtained from Algorithm 3, $\beta_1 = \beta_{11}$, $\beta_{1t} \leq \beta_1$ for all $t \in [T]$ and $\beta_1/\sqrt{\beta_2} < 1$. Suppose $\eta_l(t+1) \geq \eta_l(t) > 0$, $\eta_u(t+1) \leq \eta_u(t)$, $\eta_l(t) \to \alpha^*$ as $t \to \infty$, $\eta_u(t) \to \alpha^*$ as $t \to \infty$, $L_\infty = \eta_l(1)$ and $R_\infty = \eta_u(1)$. Assume that $\|x - y\|_\infty \leq D_\infty$ for all $x, y \in \mathcal{F}$ and $\|\nabla f_t(x)\| \leq G_2$ for all $t \in [T]$ and $x \in \mathcal{F}$. For $x_t$ generated using the ADABOUND algorithm, we have the following bound on the regret*

$$
R_T \leq \frac{D_\infty^2 \sqrt{T}}{2(1-\beta_1)} \sum_{i=1}^{d} \eta_{T,i}^{-1} + \frac{D_\infty^2}{2(1-\beta_1)} \sum_{t=1}^{T}\sum_{i=1}^{d} \beta_{1t}\eta_{t,i}^{-1} + (2\sqrt{T} - 1)\frac{R_\infty G_2^2}{1-\beta_1}.
$$

---

**Algorithm 3** AMSBOUND

---

**Input:** $x_1 \in \mathcal{F}$, initial step size $\alpha$, $\{\beta_{1t}\}_{t=1}^T$, $\beta_2$, lower bound function $\eta_l$, upper bound function $\eta_u$

1: Set $m_0 = 0$, $v_0 = 0$ and $\hat{v}_0 = 0$
2: **for** $t = 1$ **to** $T$ **do**
3:      $g_t = \nabla f_t(x_t)$
4:      $m_t = \beta_{1t}m_{t-1} + (1 - \beta_{1t})g_t$
5:      $v_t = \beta_2 v_{t-1} + (1 - \beta_2)g_t^2$
6:      $\hat{v}_t = \max(\hat{v}_{t-1}, v_t)$ and $V_t = \text{diag}(\hat{v}_t)$
7:      $\eta = \text{Clip}(\alpha/\sqrt{V_t}, \eta_l(t), \eta_u(t))$ and $\eta_t = \eta/\sqrt{t}$
8:      $x_{t+1} = \Pi_{\mathcal{F}, \text{diag}(\eta_t^{-1})}(x_t - \eta_t \odot m_t)$
9: **end for**

---

The regret of AMSBOUND has the same upper bound with that of ADABOUND.[3]

## G    EMPIRICAL STUDY ON BOUND FUNCTIONS

Here we provide an empirical study on different kinds of bound functions. We consider the following two key factors of the bound function: convergence speed and convergence target. The former one affects how "fast" our algorithms transform from adaptive methods to SGD(M), while the latter one reflects the final step size of SGD(M). In particular, we consider the following bound functions:

$$\eta_l(t) = (1 - \frac{1}{(1 - \beta)t + 1})\alpha^*,$$

$$\eta_u(t) = (1 + \frac{1}{(1 - \beta)t})\alpha^*,$$

where the above functions will converge to $\alpha^*$ and the larger $\beta$ results in lower convergence speed.

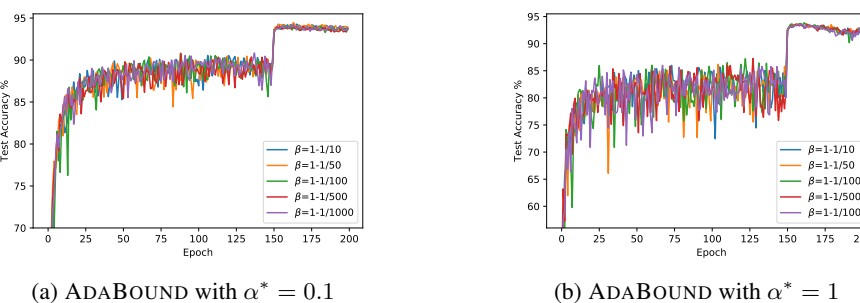

(a) ADABOUND with $\alpha^* = 0.1$            (b) ADABOUND with $\alpha^* = 1$

Figure 5: Test accuracy of ADABOUND with different $\beta$ using ResNet-34 on CIFAR-10.

We first investigate the impact of convergence speed. We conduct an experiment of ADABOUND on CIFAR-10 dataset with the ResNet-34 model, where $\beta$ is chosen in $\{1 - \frac{1}{10}, 1 - \frac{1}{50}, 1 - \frac{1}{100}, 1 - \frac{1}{500}, 1 - \frac{1}{1000}\}$ and $\alpha^*$ is chosen from $\{1, 0.1\}$. The results are shown in Figure 5. We can see that for a specific $\alpha^*$, the performances with different $\beta$ are almost the same . It indicates that the convergence speed of bound functions does not affect the final result to some extent. We find a $\beta$ in $[\beta_1, \beta_2]$ usually contributes to a strong performance across all models.

Next, we investigate the impact of convergence target and the results are displayed in Figure 6. We test SGDM and ADABOUND with different $\alpha$ (or $\alpha^*$) with the ResNet-34 model, where $\alpha$ (or $\alpha^*$) is chosen in $\{1, 0.1, 0.03, 0.01, 0.003, 0.001\}$ and $\beta = 0.99$. The results show that SGDM is very sensitive to the hyperparameter. The best value of the step size for SGDM is $0.1$ and it has large performance gaps compared with other settings. In contrast, ADABOUND has stable performance in different final step sizes, which illustrates that it is not sensitive to the convergence target.

---

[3]One may refer to Appendix E as the process of the proof is almost the same.

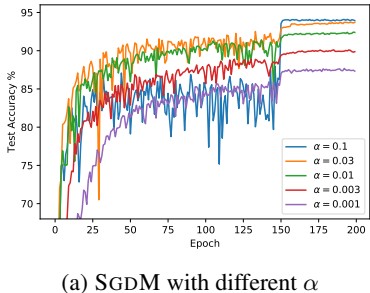

(a) SGDM with different $\alpha$

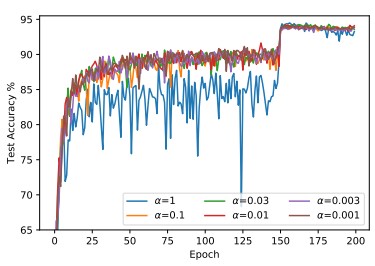

(b) ADABOUND with different $\alpha^*$

Figure 6: Test accuracy of SGDM/ADABOUND with different $\alpha/\alpha^*$ using ResNet-34 on CIFAR-10. The result of SGDM with $\alpha = 1$ is not shown above as its performance is too poor (lower than 70%) to be plotted together with other results in a single figure.

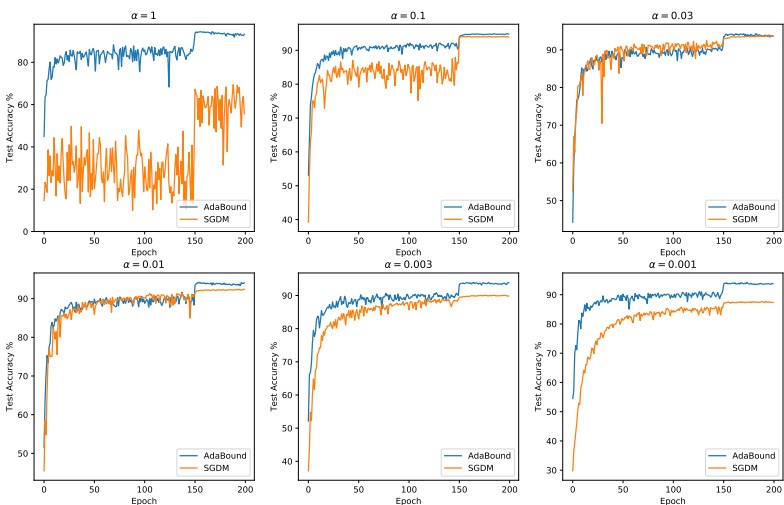

Figure 7: Comparison of test accuracy between SGDM and ADABOUND with different $\alpha/\alpha^*$.

We further directly compare the performance between SGDM and ADABOUND with each $\alpha$ (or $\alpha^*$). The results are shown in Figure 7. We can see that ADABOUND outperforms SGDM for all the step sizes. Since the form of bound functions has minor impact on the performance of ADABOUND, it is likely to beat SGDM even without carefully tuning the hyperparameters.

To summarize, the form of bound functions does not much influence the final performance of the methods. In other words, ADABOUND is not sensitive to its hyperparameters. Moreover, it can achieve a higher or similar performance to SGDM even if it is not carefully fine-tuned. Therefore, we can expect a better performance by using ADABOUND regardless of the choice of bound functions.

## H    EMPIRICAL STUDY ON THE EVOLUTION OF LEARNING RATES OVER TIME

Here we provide an empirical study on the evolution of learning rates of ADABOUND over time. We conduct an experiment using ResNet-34 model on CIFAR-10 dataset with the same settings in Section 5. We randomly choose two layers in the network. For each layer, the learning rates of its parameters are recorded at each time step. We pick the min/median/max values of the learning rates in each layer and plot them against epochs in Figure 8.

We can see that the learning rates increase rapidly in the early stage of training, then after a few epochs its max/median values gradually decrease over time, and finally converge to the final step size. The increasing at the beginning is due to the property of the exponential moving average of $\phi_t$ of ADAM, while the gradually decreasing indicates the transition from ADAM to SGD.

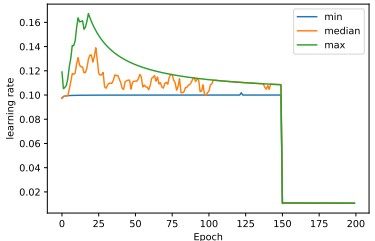 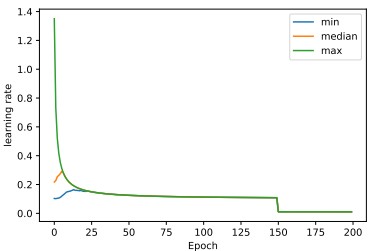

Figure 8: The evolution of learning rates over time in two randomly chosen layers.

