# OpenReview forum: "Adaptive Gradient Methods with Dynamic Bound of Learning Rate"
_ICLR.cc/2019/Conference_

### Official Review · AnonReviewer1 · 2018-10-26
**Nice experiments, but theory does not reflect any benefit**

**Rating:** 6
**Confidence:** 4

**Review:**

*Summary :
The paper explores variants of popular adaptive optimization methods.
The idea is to clip the magnitude of the gradients from above and below in order to prevent too aggressive/conservative updates.
The authors provide regret bound to this algorithm in the online convex setting and perform several illustrative experiments.


*Significance:
-There is not much novelty in Theorems 1,2,3 since similar results already appeared in Reddi et al.

-Also, the theoretical part does not demonstrate the benefit of the clipping idea. Concretely, the regret bounds seem to be similar to the bounds of AMSBound.
Ideally, I would like to see an analysis that discusses a situation where AdaGrad/AMSBound fail or perfrom really bad, yet the clipped versions do well.

-The experimental part on the other hand is impressive, and the results illustrate the usefulness of the clipping idea.

*Clarity:
The idea and motivation are very clear and so are the experiments.


*Presentation:
The presentation is mostly good.

Summary of review:
The paper suggests a simple idea to avoid extreme behaviour of the learning rate in standard adaptive methods. The theory is not so satisfying, since it does not illustrate the benefit of the method over standard adaptive methods. The experiments are more thorough and illustrate the applicability of the method.

---

> ### Author Response · Authors · 2018-11-25
> **Rebuttal**
>
>
> Thanks for your comments.
>
> >>> There is not much novelty in Theorems 1,2,3 since similar results already appeared in Reddi et al.
>
> We argue that Reddi et al. (2018) did not prove 「for all the initial learning rates」, Adam has bad behavior, and this condition is important for showing the necessity of our idea of restricting the actual learning rates. That’s why we complete the proof with this weaker assumption. We would not claim the theoretical analysis as our main contribution in this paper, but it is a necessary part that serves for our actual main contribution「proposing the idea of an optimization algorithm that can gradually transform from adaptive methods to SGD(M), combining both of their advantages」. All the other parts in the paper, including preliminary empirical study, theoretical proofs, experiments, and further analysis, serve for this main contribution.
>
> >>> Also, the theoretical part does not demonstrate the benefit of the clipping idea. Concretely, the regret bounds seem to be similar to the bounds of AMSBound. Ideally, I would like to see an analysis that discusses a situation where AdaGrad/AMSBound fail or perform really bad, yet the clipped versions do well.
>
> First, the name of our new proposed methods are AdaBound and AMSBound. I guess you mean AMSGrad in your suggestion?
> Actually, it is easy to use a setting similar to that of Wilson et al. (2017), to show AdaGrad/Adam achieve really bad performance while our methods do well. But I don’t think it is very meaningful since it is only a bunch of examples. As also mentioned by review 2, the average performance of the algorithms is what really matters. But due to its difficulty, most similar works on optimizers tend to use experiments to support their arguments and lack the theoretical proofs for this part.

---

> > ### Comment · AnonReviewer1 · 2018-12-03
> > **Comment**
> >
> > I thank the reviewers for their response, and I keep my score.

---

### Official Review · AnonReviewer3 · 2018-11-01

**Rating:** 4
**Confidence:** 5

**Review:**

The authors introduce AdaBound, a method that starts off as Adam but eventually transitions to SGD. The motivation is to benefit from the rapid training process of Adam in the beginning and the improved convergence of SGD at the end. The authors do so by clipping the weight updates of Adam in a dynamic way. They show numerical results and theoretical guarantees. The numerical results are presented on CIFAR-10 and PTB while the theoretical results are shown on assumptions similar to AMSGrad (& using similar proof strategies). As it stands, I have some foundational concerns about the paper and believe that it needs significant improvement before it can be published. I request the authors to please let me know if I misunderstood any aspect of the algorithm, I will adjust my rating promptly. I detail my key criticisms below:

- I'm somewhat confused by the formulation of \eta_u and \eta_l. The way it is set up (end of Section 4), the final learning rate for the algorithm converges to 0.1 as t goes to infinity. In the Appendix, the authors show results also with final convergence to 1. Are the results coincidental with the fact that SGD works well with those learning rates? It is a bit odd that we indirectly encode the final learning rate of the algorithm into the \eta s.

- Am I correct in saying that with t=100 (i.e., the 100th iteration), the \eta s constrain the learning rates to be in a tight bound around 0.1? If beta=0.9, then \eta_l(1) = 0.1 - 0.1 / (0.1*100+1) = 0.091. After t=1000 iterations, \eta_l becomes 0.099. Again, are the good results coincidental with the fact that SGD with learning rate 0.1 works well for this setup? In the scheme of the 200 epochs of training (equaling almost 100-150k iterations), if \eta s are almost 0.099 / 0.10099, for over 99% of the training, we're only doing SGD with learning rate 0.1.

- Along the same lines, what learning rates on the grid were chosen for each of the problems? Does the setup still work if SGD needs a small step size and we still have \eta converge to 1? A VGG-11 without batch normalization typically needs a smaller learning rate than usual; could you try the algorithms on that?

- Can the authors plot the evolution of learning rate of the algorithm over time? You could pick the min/median/max of the learning rates and plot them against epochs in the same way as accuracy.This would be a good meta-result to show how gradual the transition from Adam to SGD is.

- The core observation of extreme learning rates and the proposal of clipping the updates is not novel; Keskar and Socher (which the authors cite for other claims) motivates their setup with the same idea (Section 2 of their paper). I feel that the authors should clarify what they are proposing as novel. Is it correct that a careful theoretical analysis of this framework is what stands as the authors' major contribution?

- Can you try experimenting with/suggesting trajectories for \eta which converge to SGD stepsize more slower?

- Similarly, can you suggest ways to automate the choice for the \eta^\star? It seems that the 0.1 in the numerator is an additional hyperparameter that still might need tuning?

---

> ### Author Response · Authors · 2018-11-25
> **Rebuttal: about bound functions**
>
>
> Thanks for your questions and suggestions. We separate the questions into 3 parts (bound functions, contributions, and extra details & experiments) and post the responses below. We hope they can address your questions.
>
> [About bound functions]
>
> We want to clarify the following facts about the bound function:
> 1. The convergence speed (indicated by \beta in current settings) and convergence target (indicated by \alpha*) exert minor impacts on the performance of AdaBound.
> 2. In other words, AdaBound is not sensitive to the form of bound functions, and therefore we don’t have to waste much time fine-tuning the hyperparameters, especially compared with SGD(M).
> 3. Moreover, even not carefully fine-tuned AdaBound can beat SGD(M) with the optimal step size.
>
> We conducted the empirical study in Appendix G in order to illustrate the above points. But as you have raised a few questions about the bound function, it seems that our original experiments are not enough. We expand the experiments in an attempt to give more evidence to support the above statements and hope this can answer some questions you mentioned.
>
> >>> I'm somewhat confused by the formulation of \eta_u and \eta_l. The way it is set up (end of Section 4), the final learning rate for the algorithm converges to 0.1 as t goes to infinity. In the Appendix, the authors show results also with final convergence to 1. Are the results coincidental with the fact that SGD works well with those learning rates? It is a bit odd that we indirectly encode the final learning rate of the algorithm into the \eta s.
>
> (Note: SGD and SGDM have similar performance in our experiments. Here we directly use SGD to generally indicate SGD or SGDM)
> It is not a coincidence. SGD is very sensitive to the step size. \alpha=0.1 is the best setting and other settings have large performance gaps compared with the optimal one (see Figure 6a). But AdaBound has stable performance in different final step sizes (see Figure 6b). Moreover, for all the step sizes, AdaBound outperforms SGD (see Figure 7).
>
> >>> Can you try experimenting with/suggesting trajectories for \eta which converge to SGD stepsize more slower?
>
> We further test \beta for {1-1/10, 1-1/50, 1-1/100, 1-1/500, 1-1/1000}, which translates to some slower convergence speed of bound functions. Their performances are really close (see Figure 5).
>
> >>> Similarly, can you suggest ways to automate the choice for the \eta^\star? It seems that the 0.1 in the numerator is an additional hyperparameter that still might need tuning?
>
> In the current form of bound functions, yes, it is an additional hyperparameter. But as illustrated by the experiments, AdaBound is very robust and not sensitive to hyperparameters (we can randomly use \alpha from 0.001 to 1 and still get stable and good performance). I think in practice, we can somehow treat it as “no need of tuning”, and 0.1 can be a default setting.

---

> > ### Author Response · Authors · 2018-11-25
> > **about contributions**
> >
> >
> > [About contributions]
> >
> > >>> Is it correct that a careful theoretical analysis of this framework is what stands as the authors' major contribution?
> >
> > We want to clarify that our main contribution is: 「proposing the idea of an optimization algorithm that can gradually transform from adaptive methods to SGD(M), combining both of their advantages」
> > All the other parts in the paper, including preliminary empirical study, theoretical proofs, experiments, and further analysis, serve for the main contribution. From Wilson et al. (2017), many researchers have been devoted to finding a way to train as fast as Adam and as good as SGD. Many of them failed and some of them present so complicated algorithms.
> > The purpose of this paper is to tell other researchers that such an interesting, simple and direct approach can achieve surprisingly good and robust performance. Note that “bound functions on learning rates” is only one particular way to conduct “gradual transformation from Adam to SGD”. There might be other ways that can work too, such as well-designed decay. We think publicizing now with several baseline experiments and a basic theoretical proof so as to stimulate other people's research can benefit the research community.
> >
> > >>> The core observation of extreme learning rates and the proposal of clipping the updates is not novel;
> >
> > We are not the first to propose clipping of learning rates. But we would argue that no one has given a clear observation of the existence of extreme learning rates before. Wilson et al. (2017) first mentioned that extreme learning rates may cause bad performance, but it is just an assumption. Keskar & Socher (2017)’s preliminary experiment can be seen as indirect evidence. As far as we know, we are the first that directly show both extremely large and small learning rates exist in the final stage of training.
> >
> > >>> Keskar and Socher (which the authors cite for other claims) motivates their setup with the same idea (Section 2 of their paper). I feel that the authors should clarify what they are proposing as novel.
> >
> > We will clarify that the idea of learning rate clipping has been proposed by Keskar & Socher (2017).
> > Even if they had not mentioned the idea of clipping learning rates, we wouldn’t claim it as our new contribution. Actually, clipping is really common in practice/in many frameworks’ API. The difference is that we usually use it on gradients. We have also mentioned the above facts in Section 4.
> > Also, we want to clarify again that our main contribution is the idea of “gradual transformation from Adam to SGD”, and clipping is just one particular way of implementation.
> > It should also be mentioned that this part in Keskar & Socher (2017) is preliminary. They did not give a thorough discussion about clipping or extreme learning rates.

---

> > > ### Author Response · Authors · 2018-11-25
> > > **about extra details & experiments**
> > >
> > >
> > > [About details and extra experiments you asked for]
> > >
> > > >>> Am I correct in saying that with t=100 (i.e., the 100th iteration), the \eta s constrain the learning rates to be in a tight bound around 0.1? If beta=0.9, then \eta_l(1) = 0.1 - 0.1 / (0.1*100+1) = 0.091. After t=1000 iterations, \eta_l becomes 0.099. Again, are the good results coincidental with the fact that SGD with learning rate 0.1 works well for this setup? In the scheme of the 200 epochs of training (equaling almost 100-150k iterations), if \eta s are almost 0.099 / 0.10099, for over 99% of the training, we're only doing SGD with learning rate 0.1.
> > >
> > > Actually, we used \beta_1=0.99 in our experiments. Therefore \eta_l comes to 0.091 at t=1000 rather than t=100, and it is about 10 epochs. Also, as mentioned above, we test larger \beta and the performance are similar (see Figure 5).
> > >
> > > >>> Along the same lines, what learning rates on the grid were chosen for each of the problems?
> > >
> > > The settings are:
> > > SGD(M): \alpha=0.1 for MNIST/CIFAR10, \alpha=10 for PTB; momentum=0.9
> > > Adam, AMSBound: \alpha=0.001, \beta_1=0.99, \beta_2=0.999
> > > AdaGrad: \alpha=0.01
> > >
> > > We only provided the grid search sets of hyperparameters due to the page limit before.
> > > We will soon add a section in the appendix to illustrate the specific settings of hyperparameters for all the optimizers.
> > >
> > > >>> Does the setup still work if SGD needs a small step size and we still have \eta converge to 1? A VGG-11 without batch normalization typically needs a smaller learning rate than usual; could you try the algorithms on that?
> > >
> > > Yes. We add an experiment according to your suggestion (VGG-11 without batch normalization on CIFAR-10, using AdaBound/AMSBound, SGD, and other baselines). The best step size for SGD is 0.01 and AdaBound with \alpha*=1 still have similar performance with the best-tuned SGD (see this anonymous link to the results: https://github.com/AgentAnonymous/X/blob/master/vgg_test.pdf ) .
> > >
> > > >>> Can the authors plot the evolution of learning rate of the algorithm over time? You could pick the min/median/max of the learning rates and plot them against epochs in the same way as accuracy. This would be a good meta-result to show how gradual the transition from Adam to SGD is.
> > >
> > > We conduct an experiment as you suggested, the results are placed in Appendix H.
> > > For short, we can see that the learning rates increase rapidly in the early stage of training, then after a few epochs its max/median values gradually decrease over time, and finally converge to the final step size. The increasing at the beginning is due to the property of the exponential moving average of \phi_t of Adam, while the gradually decreasing indicates the transition from Adam to SGD.

---

### Official Review · AnonReviewer2 · 2018-11-02
**Review of "Adaptive Gradient Methods with Dynamic Bound of Learning Rate"**

**Rating:** 7
**Confidence:** 4

**Review:**

This paper presents new variants of ADAM and AMSGrad that bound the gradients above and below to avoid potential negative effects on generalization of excessively large and small gradients; and the paper demonstrates the effectiveness on a few commonly used machine learning test cases.  The paper also presents detailed proofs that there exists a convex optimization problem for which the ADAM regret does not converge to zero.

This paper is very well written and easy to read.  For that I thank the authors for their hard word.  I also believe that their approach to bound is well structured in that it converges to SGD in the infinite limit and allows the algorithm to get teh best of both worlds - faster convergence and better generalization.  The authors' experimental results support the value of their proposed algorithms.  In sum, this is an important result that I believe will be of interest to a wide audience at ICLR.

The proofs in the paper, although impressive, are not very compelling for the point that the authors want to get across.  That fact that such cases of poor performance can exists, says nothing about the average performance of the algorithms, which is practice is what really matters.

The paper could be improved by including more and larger data sets.  For example, the authors ran on CIFAR-10.  They could have done CIFAR-100, for example, to get more believable results.

The authors add a useful section on notation, but go on to abuse it a bit.  This could be improved.  Specifically, they use an "i" subscript to indicate the i-th coordinate of a vector and then in the Table 1 sum over t using i as a subscript.  Also, superscript on vectors are said to element-wise powers.  If so, why is a diag() operation required?  Either make the outproduct explicit, or get rid of the diag().

---

> ### Author Response · Authors · 2018-11-25
> **Rebuttal**
>
>
> Thanks for your comments!
>
> We deeply agree that the average performance of different algorithms is very important in practice. But as also mentioned in the reply to anonymous comments before (on 11.12), our understanding of the generalization behavior of deep neural networks is still very shallow by now. It is a big challenge of investigating from theoretical aspects. Actually, the theoretical analysis of most recent related work is still under strong or particular assumptions. I believe if one could conduct convincing theoretical proof without strong assumptions, that work is totally worth an individual publication.
>
> We are conducting more experiments on larger datasets such as CIFAR-100 and on more tasks in other fields, and the results are very positive too. We will add the results and analysis in the final revision if there is space left in the paper.
>
> We want to argue that the use of diag() is necessary since \phi_t is a matrix rather than a vector. Also, $g$ is not a vector but $g_t$ is, and $g_{t,i}$ is coordinate.
> It is true that the expression $x_i$ might be ambiguous without context: 1) $x$ is a vector and it means the i-th coordinate of $x$ or 2) $x$ is not a vector and $x_i$ is a vector at time $i$. But since $x$ cannot be or not be a vector at the same time, it is clear in a specific context. This kind of notation is also used in many other works. We re-check the math expressions in our paper and think they are ok.

---

### Public Comment · (anonymous) · 2018-10-09
**About the code**

Hi!  I am interested in the algorithm you proposed and want to have a try on my researches. Could you provide an implementation of the algorithm? Or, if it is not convenient in the review period, could you give a brief instruction of how to implement it?
Good luck and hope your paper can be accepted. :-)

---

> ### Author Response · Authors · 2018-10-09
> **Thank you for your interests**
>
> Thank you for your interests!
>
> Honestly, the code is a little bit messy currently. We are cleaning up the code for releasing it these days.
> If you can't wait to have a try, it is easy to implement the algorithm by making some minor changes on the optimizers in PyTorch. Take AdaBound/AMSBound as an example, we just modify the source code of Adam (https://github.com/pytorch/pytorch/blob/master/torch/optim/adam.py). Specifically, we use torch.clamp(x, l, r) function, which can constrain x between l and r element-wisely, to perform the clip operation mentioned in the paper. You can also make similar changes to other optimizers such as AdaDelta and RMSprop.
>
> The codes for the experiments in the paper, as mentioned in the footnote on page 6, are obtained from https://github.com/kuangliu/pytorch-cifar and https://github.com/salesforce/awd-lstm-lm.
>
> We would be happy if you can share your results on your own researches using our methods.

---

### Public Comment · ~Hyesst_Wu1 · 2018-10-10
**Some questions**

Hi, thanks for the nice paper. The way of combining the adaptive methods and SGD proposed in the paper is really interesting, while I guess I find some small typos or mistakes. They are all minor and do not much affect the understand of the paper, but I think a clarification on them would be fine.

First, the upper bound function at the end of Section 4 and Appendix G does not converge to 0.1. I believe it is a typo: there is a redundant "1" at the denominator and the correct expression should be $0.1 + \frac{0.1}{(1-\beta)t}$. Also, I guess you miss the subscripts of $\beta$ in the functions in Section 4. Maybe it should be $\beta_1$ or $\beta_2$, I guess.

Second, how many layers do you use in DenseNet? You provide the source code you used for DenseNet, and it is DenseNet-121 in the code. However, I suggest mentioning the number of layers directly in the paper. It is an important hyperparameter of deep CNN network.

---

> ### Author Response · Authors · 2018-10-11
> **Clarification**
>
> Hi Hyesst,
>
> Thanks for your interests.
>
> 1. You are absolutely right! Thanks for your correction! It should be $\beta_1$ in the upper bound function at the end of Section 4.
>
> 2. Yes, we used DenseNet-121. We will add this information in the next revision.
>
> Thank you very much for your comments and suggestions.

---

### Public Comment · (anonymous) · 2018-10-11
**Clarification about ADAGRAD and generalization of your method**

Hi,

i have three main questions for you. It would be great if you could help clarify them.

1. You mention  the following about ADAGRAD along with ADAM and RMSPROP - "they are observed to generalize poorly compared with SGD or even fail to converge due to unstable and extreme learning rates.". As far as I am aware the issue with the convergence analysis of exponentiated squared gradient averaging algorithms like ADAM and RMSPROP do not extend to ADAGRAD. So, ADAGRAD is indeed guaranteed to converge given the right assumptions. In the rest of the paper, the experiments and arguments mainly consist of ADAM and not adaptive methods in general. So I think the distinction between adaptive methods in general and adaptive methods like ADAM and RMSPROP with respect to convergence guarantees should be made clearer.

2. I am not sure I understand but could you please clarify how AMSGRAD helps in the generalization of ADAM. From my understanding, it only solved the convergence issue by ensuring that the problematic quantity in the proof is PSD.

3. You mention "Experimental results show that new variants can eliminate the generalization gap
between adaptive methods and SGD". Given that the paper only contains a few empirical results (on some important and common tasks) and no theoretical proof in that respect, I find it to be a misleading statement. The experiments in Wilson et. al(2017) give proper evidence of the gap between SGD and Adaptive methods in overparameterized settings. To show that this method overcomes it, I think you need a stronger argument than what you have shown.

---

> ### Author Response · Authors · 2018-10-12
> **Part1 of Clarifications**
>
> Thanks for your interests.
>
> I respond point by point below.
>
> >>> "they are observed to generalize poorly compared with SGD or even fail to converge due to unstable and extreme learning rates." As far as I am aware the issue with the convergence analysis of exponentiated squared gradient averaging algorithms like ADAM and RMSPROP do not extend to ADAGRAD. So, ADAGRAD is indeed guaranteed to converge given the right assumptions.
>
> A more precise expression of that sentence should be "Adam, RMSprop, AdaGrad, and other adaptive methods are observed to generalize poorly ..., and, some of them (i.e. Adam) even fail to converge ...", which is summarized from [1][2][3] and Section 3 in our paper. We didn't notice the original sentence might be misunderstood. We will use a more precise way to summarize the phenomenon. However, although AdaGrad is theoretically guaranteed to converge, it is well-accepted that in practice the convergence is too slow at the end of training due to its accumulation of second order momentum. As we usually use a limited number of epochs or limited time in a training job, it may fail to achieve the "theoretical convergence". Therefore, maybe we can say "may hard to converge" to summarize. :D
>
> >>> In the rest of the paper, the experiments and arguments mainly consist of ADAM and not adaptive methods in general. So I think the distinction between adaptive methods in general and adaptive methods like ADAM and RMSPROP with respect to convergence guarantees should be made clearer.
>
> The main purpose of this paper is to introduce a novel framework that can combine the advantages of adaptive methods and SGD(M). The framework applies to Adam as well as AdaGrad and other adaptive methods. As mentioned above, the weaknesses of adaptive methods are in common and combining with SGD can help overcome the problems. Therefore, we don't think it is necessary to distinguish particular adaptive methods everywhere in the paper. We run experiments mainly on Adam because of its popularity. According to your comments, we would consider adding more experiments on other adaptive methods like AdaGrad.
>
> >>> I am not sure I understand but could you please clarify how AMSGRAD helps in the generalization of ADAM. From my understanding, it only solved the convergence issue by ensuring that the problematic quantity in the proof is PSD.
>
> I guess we understand "generalization" differently. If you regard "generalization error", in a narrow sense, as how large the gap between training and testing error is, then I agree that AMSGrad only solves the convergence issue. But broadly speaking, "generalization error" is a measure of how accurate of a method on unseen data (see https://en.wikipedia.org/wiki/Generalization_error). It depends on not only handling overfitting but the convergence results on training data. Therefore, attempts on solving convergence issue can also help the generalization in a broad sense.
>
> >>> The experiments in Wilson et. al(2017) give proper evidence of the gap between SGD and Adaptive methods in overparameterized settings. To show that this method overcomes it, I think you need a stronger argument than what you have shown.
>
> We would first argue that the experiments in Wilson et al. (2017), including a few common tasks in CV and NLP, are not much different to ours and that in other recent similiar works. While, their artifactual example before the experiment section does use a overparameterized settings, but they never claim it is the main cause of poor generalization. It is a necessary but not sufficient condition. Indeed, the poor generalizaion is mainly caused by the propery of the carefully constructed particular task. In other words, it is highly problem-dependent. The actual statement of Wilson et al. (2017) is
>
> ** When a problem has multiple global minima, different algorithms can find entirely different solutions when initialized from the same point. In addition, we construct an example where adaptive gradient methods find a solution which has worse out-of-sample error than SGD. **
>
> Therefore, no one can affirm there are no examples that adaptive methods find a better solution than SGD. The above are just examples and there are infinite exmaples. We don't think it is meaningful to show our framework can perform well on that particular one, even though it is not hard.

---

> > ### Author Response · Authors · 2018-10-12
> > **Part2**
> >
> > >>> You mention "Experimental results show that new variants can eliminate the generalization gap between adaptive methods and SGD". Given that the paper only contains a few empirical results (on some important and common tasks) and no theoretical proof in that respect, I find it to be a misleading statement.
> >
> > Honestly, maybe I don't exactly get your point. We said "experimental results show", not "theoretical proof show" or something like that. If you said "your experiments are not enough", I could understand and may add some additional experiments on other tasks if reasonable. But we don't think "no theoretical proof in that respect" is a valid point to criticize that the statement is misleading or overclaiming.
> >
> > In addition, it should be mentioned that our understanding of the generalization behavior of deep neural networks is still very shallow by now. It is a big challenge of investigating from theoretical aspects. A summary of recent achievements can be found here (http://ruder.io/deep-learning-optimization-2017/), and we can see their theoretical analysis is still under strong or particular assumptions. That's why most similar works on optimizers tend to use experiments to support their arguments.
> >
> > As for the richness of our experiments, in our paper the tasks include several popular ones on CV and NLP area; the models include simple perceptron, deep CNN models and RNN models. We give a brief comparison to some recent works for a fair judgment.
> >
> > - [1] does not propose novel algorithms or frameworks as we do. Their main contribution is empirically showing that the minima found by adaptive learning rate methods perform generally worse compared to those found by SGD, and providing some possible causes. The richness of experiments of ours is similar to theirs. Personally, the amount of experiments in this work is an average level among similar works as far as I know.
> > - The experiments in [2] are very limited, as the authors also state the experiments are "preliminary".
> > - [3] conducts more experiments than other similar works. But there is no theoretical analysis, which is important in such kinds of works.
> > - [4] (posted on arXiv and also a submission to ICLR19) only conducts experiments on image classification tasks. As it is known that the gap between Adam and SGD on this task is notable, while on some NLP tasks like machine translation, well-turned Adam may even outperform SGD ([6]), it is not enough to just test on this single task.
> > - The experiments in [5] (posted on arXiv and also a submission to ICLR19) are even more limited than that of [2], only a toy model on MINST.
> >
> > Therefore, we argue that our experiments have already shown the potential of our proposed framework. Future papers by other researchers are a more appropriate home for additional experiments on other tasks. We think publicizing now with the set of baselines that we have already included so as to stimulate others' research is more effective than us delaying publication and presentation of this work.
> >
> > -----
> > [1] Wilson, A.C., Roelofs, R., Stern, M., Srebro, N., & Recht, B. (2017). The Marginal Value of Adaptive Gradient Methods in Machine Learning. NIPS.
> > [2] Sashank J.R., Satyen K., & Sanjiv K. (2018). On the Convergence of Adam and Beyond. ICLR.
> > [3] Keskar, N.S., & Socher, R. (2017). Improving Generalization Performance by Switching from Adam to SGD. CoRR, abs/1712.07628.
> > [4] Chen, J., & Gu, Q. (2018). Closing the Generalization Gap of Adaptive Gradient Methods in Training Deep Neural Networks. CoRR, abs/1806.06763.
> > [5] Chen, X., Liu, S., Sun, R., & Hong, M. (2018). On the Convergence of A Class of Adam-Type Algorithms for Non-Convex Optimization. CoRR, abs/1808.02941.
> > [6] Denkowski, M.J., & Neubig, G. (2017). Stronger Baselines for Trustable Results in Neural Machine Translation. NMT@ACL.

---

### Public Comment · (anonymous) · 2018-10-16
**A few comments**

Dear authors,

Interesting work! My coauthors and I have been suffered from the poor generalization of Adam in many of our productions for a long time. We have to use SGD for better performance but I do HATE fine-tuning hyperparameters of SGD again and again!

I noticed that there have been many new proposed optimizers claiming they are better than Adam. I once tried some of them and was disappointed to find that they can bring nothing improvement but more hyperparameters! I doubt that the more and more complicated design of optimizers is not a right way and there must be a simple way to build an optimizer as fast as Adam while as good as SGD.

That’s why this paper really attracts me. The idea of gradually transforming Adam to SGD is really simple but looks intuitive and reasonable. It makes sense to me. The algorithm is also well-presented. I am surprised that you also provide convincing proofs about the algorithm --- I had thought you would just construct some empirical studies w/o theoretical analysis.

I have a few questions about the paper and personal thoughts of the future work. I hope they will be useful to the authors. Feel free to leave them as is if they are not correct. :D

- Besides rapid training, the smoothness of the learning curve is another advantage of adaptive methods. Personally, I think it might be more important. When trying to train new models, we often do not know whether it can converge in advance. A common approach is training few epochs and making a preliminary decision of what to do next based on the trend of learning curve in the early stage. The sharp fluctuation of loss is common when using SGD, which makes it hard to estimate the trend of learning curve quickly. Are your framework able to keep this strength of Adam? What is your take on this?

- I tried AdaBound on CIFAR-10 by myself. It is interesting that I have used simpler bound functions (linear functions and piecewise constant function) and still got very good performance. As you also mentioned that the convergence speed of bound functions is not very important, I suggest you may choose simpler ones (Occam’s Razor).

- I am thinking about if we may use a way like in Keskar et al. (2017) to determine the final step size automatically. I didn’t think through this carefully of whether it is possible. What is your opinions?

Thanks in advance for your time and I hope this paper get accepted!

---

> ### Author Response · Authors · 2018-10-25
> **Some answers and responses**
>
> Sorry for the late response. It's been a bit busy in the past few days. Thanks for your comments and we present our responses below.
>
> 1. We didn't pay much attention to the smoothness of the learning curve before and the analysis mainly focuses on training speed w.r.t. the advantage of adaptive methods. But, actually, we also mentioned that the learning curve of our framework is smoother than that of SGD in the experiment on PTB (in para. 1, section 5.4, page 8). We would agree with your opinion that the smoothness is also important. We are pleased to add some more discussion on this point in the next revision.
>
> 2. That's interesting. It is a good engineering question that what particular bound function is best (simplicity, efficiency, effectiveness etc.) in production. As for this paper, it is more about to show the potential of a novel framework and stimulate others' research. It would be a direction of future work to investigate whether there is a simpler bound function that guarantees the performance.
>
> 3. I am afraid that the method in Keskar et al. (2017) seems not able to be applied to our algorithm directly. Introducing automation is meaningful. But it is not a very easy task, IMO. We may think about this point carefully in the future work.

---

### Public Comment · (anonymous) · 2019-10-09
**proof of bounds on the effective stepsize**

Assuming epsilon = 0,the effective step taken in parameter space at timestep t is ∆t = αt · m t/√vt.
"The effective stepsize has
two upper bounds:
|∆t| ≤ α · (1 − β1)/√1 − β2 in the case (1 − β1) >√1 − β2, and
|∆t| ≤ α otherwise"
How do you prove this? since α^t = α ·√(1 − β2^t)/(1 − β1^t) can go to positive infinity too

---

### Meta-Review · Area_Chair1 · 2018-12-12
**Summary review**

**Confidence:** 5
**Recommendation:** Accept (Poster)

**Metareview:**

The paper was found to be well-written and conveys interesting idea. However the AC notices a large body of clarifications that were provided to the reviewers (regarding the theory, experiments, and setting in general) that need to be well addressed in the paper.